# Separating the *what* and *how* of compositional computation to enable reuse and continual learning

**Haozhe Shan**[*]
Center for Theoretical Neuroscience
Department of Computer Science
Columbia University
hs3594@columbia.edu

**Sun Minni**[*]
Center for Theoretical Neuroscience
Department of Neuroscience
Columbia University
ms5724@columbia.edu

**Lea Duncker**
Center for Theoretical Neuroscience
Department of Neuroscience
Columbia University
ld3149@columbia.edu

## Abstract

The ability to continually learn, retain and deploy skills to accomplish goals is a key feature of intelligent and efficient behavior. However, the neural mechanisms facilitating the continual learning and flexible (re-)composition of skills remain elusive. Here, we study continual learning and the compositional reuse of learned computations in recurrent neural network (RNN) models using a novel two-system approach: one system that infers *what* computation to perform, and one that implements *how* to perform it. We focus on a set of compositional cognitive tasks commonly studied in neuroscience. To construct the *what* system, we first show that a large family of tasks can be systematically described by a probabilistic generative model, where compositionality stems from a shared underlying vocabulary of discrete task epochs. We develop an unsupervised online learning approach that can learn this model on a single-trial basis, building its vocabulary incrementally as it is exposed to new tasks, and inferring the latent epoch structure as a time-varying computational context within a trial. We implement the *how* system as an RNN whose low-rank components are composed according to the context inferred by the *what* system. Contextual inference facilitates the creation, learning, and reuse of low-rank RNN components as new tasks are introduced sequentially, enabling continual learning without catastrophic forgetting. Using an example task set, we demonstrate the efficacy and competitive performance of this two-system learning framework, its potential for forward and backward transfer, as well as fast compositional generalization to unseen tasks.

## 1 Introduction

Humans and animals exhibit a remarkable ability to learn and retain new skills, and flexibly deploy them to accomplish goals in an ever-changing environment. A growing literature from neuroscience and human behavior suggests that mechanisms for contextual inference and task abstraction may play a crucial role for behavioral flexibility and learning [1–9]. An abstract, task-relevant context could guide the selection and composition of different skills, while the maintenance of context-specific

---

[*]Equal contribution.

39th Conference on Neural Information Processing Systems (NeurIPS 2025).

memories could counteract forgetting and aid continual learning [1]. Yet, the neural mechanisms facilitating the continual learning and flexible reuse of skills are not well understood. How do neural systems manage to expand their repertoire of skills without interference or forgetting, while maintaining the ability to access and compose them in new environmental contexts? We hypothesize that multiple interacting learning systems with different objectives may contribute to overall learning and computation. Specifically, can inferences about the higher-level compositional structure of tasks be exploited for more robust and efficient learning?

We investigate the interplay of compositionality, contextual inference, and continual learning in recurrent neural networks (RNNs) using a two-systems approach: one that infers an abstract *computational context* within a task family by parsing the compositional structure with probabilistic inference, and one that implements and retains the relevant computation for each context. This framework naturally maps onto a *what* and *how* architecture, previously explored in meta-learning [10]. The explicit separation into contexts (what) and computations (how) allows for flexible learning and compositional reuse. We focus our analyses on a family of cognitive and motor tasks commonly used in neuroscience [11]. Here, we expand upon previous work in RNNs, where continual learning is still a major challenge [12], and compositional reuse of computational building blocks has only been shown to emerge implicitly over simultaneous training on multiple related tasks [13–16], rather than being explicitly implemented as a solution that facilitates sequential learning of tasks.

The paper is organized as follows: We review related work and our contributions in section 2. In section 3, we formalize task compositionality through the development of a generative model for cognitive tasks. This allows us to define a computational context within a task family. We develop an unsupervised online learning and inference approach to directly infer context from the inputs and target responses that constitute a task, one trial at a time across a number of different but related tasks. In section 4 we combine inference and learning in the task model with a contextually gated low-rank RNN. Training this architecture on a number of cognitive tasks, we demonstrate that our approach counteracts catastrophic forgetting and aids continual learning with signatures of both forward and backward transfer. We demonstrate competitive performance in comparison to continual learning approaches previously used in RNNs. We show that our architecture can rapidly learn to deploy known skills to solve novel tasks using only a few examples – a property called compositional generalization [17–19]. Finally, we discuss limitations and future directions in section 5.

Our work contributes to a more precise quantification of task compositionality, how it may shape computation, and how interactions across different learning systems can benefit continual learning and generalization. Ultimately, addressing these questions will help to elucidate flexibility and learning in biological systems.

## 2 Related work and our contributions

**Statistical theories of contextual inference.** We follow an influential line of work on contextual inference in cognitive sciences [1, 20–22]. Contextual inference has been proposed as a model explaining many learning-related phenomena in classical conditioning [20] and motor control [1]. Here, a latent discrete-valued random variable is used to describe distributional changes in task-relevant, continuous observations. Given a set of observations, this variable can be inferred as a familiar or novel context to determine whether a learner may express and refine existing behaviors, or expand their repertoire using a new context. We extend this idea to the problem of learning to solve multiple common neuroscience tasks [11, 14]. To do this, we develop a formal description of these tasks in terms of a context variable: Each task transitions through several contexts, each relevant for a different computation on a particular task epoch. A key aspect of our model is that these contexts can reappear in a compositional manner across different tasks, and may additionally share complex temporal dependencies within a given trial. In section 3 we show how this is crucial for capturing the distributional structure of neuroscience tasks, where, for example, the animal needs to compute a response based on information presented at an earlier time in the trial [11, 14].

**Task-dependent modulation of neural networks.** The general notion of a separate system modulating computation in a neural network in a task-dependent manner has been previously studied. An example of this is the Task-Conditioned Hypernetwork [23], where the weights of a neural network are generated by a second, separate network based on modifiable task embeddings. Other approaches often assume the network has its own learnable weights but that their involvement in computation

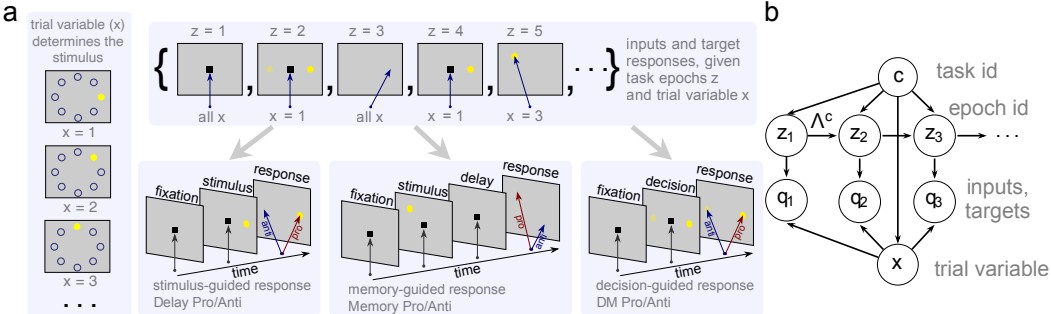

Figure 1: A generative model of a family of cognitive tasks used in neuroscience. **a**: Schematic illustration of a set of cognitive tasks and their compositional nature. **b**: Directed Acyclic Graphical model (DAG) description of the generative model capturing the distribution over cognitive tasks we consider. The time-varying inputs and target responses of each task can be modeled as a mixture, where the observations for each task are composed of a set of discrete epochs, as illustrated in (**a**).

is modulated, for instance via a gating mechanism [16, 24, 25]. These works typically study tasks that can be solved by a single subcomponent (e.g., [25–27]) or the simple linear addition of outputs from multiple subcomponents (e.g., [24, 28]). In our case, we study tasks that can only be solved via a composition of computations carried out by different subcomponents. Our choice of architecture is similar to that of Costacurta et al. [16], where an RNN receives a low-dimensional, time-varying 'neuromodulatory' signal from a second RNN that adjusts how different low-rank components of its recurrent weights are combined. While Costacurta et al. [16] train this architecture end-to-end with backpropagation, we use the interpretable output of a contextual inference algorithm as the modulatory signal. In doing so, we make the compositional reuse of RNN components across contexts explicit, thereby facilitating continual learning and compositional generalization. Beyond RNNs, the 'local module composition' scheme for feedforward networks [29] is similar to our work in spirit, except that theirs requires passing each datum to all modules in order to determine the right composition to use and does not make use of an probabilistic inference procedure.

**Continual learning in RNNs.** Continual learning is relatively understudied in RNNs and sequence-to-sequence problems more generally [12, 30]. In our comparisons, we focus on two previous continual learning approaches for networks with fixed dimensionality: Elastic weight consolidation (EWC) [31], which penalizes weight changes during future learning according to an estimate of how important each individual weight is for the performance of previous tasks. EWC is representative of a family of weight-regularization-based approaches [32]. Another approach, Orthogonal Weight-Space Projection (OWP), was developed in [13], which pushes the network to solve different computations with activity in orthogonal subspaces. This method is representative of a family of approaches that modify weight updates during learning [33–35]. We found that our approach mitigates forgetting better than these methods, exhibits forward and backward transfer learning, and can rapidly learn new tasks by re-composing learned computations.

## 3 Formalizing task compositionality via a probabilistic formulation

### 3.1 A generative model of compositional cognitive tasks

Our goal is to study how the inference of a task-relevant computational context could guide the selection and composition of different skills in an RNN and aid continual learning. To address this question, we initially need to define what a task-relevant computational context is, and how it may be inferred as part of RNN training. We do this for a family of compositionally related cognitive tasks which have been popular both in experimental and computational neuroscience [11, 13, 14, 16, 36].

In these tasks, the learner receives a sequence of time-varying inputs $\boldsymbol{s}_{t=1,\dots,T}$ and needs to produce a target response sequence $\boldsymbol{y}_{t=1,\dots,T}$, following different distributions $p(\boldsymbol{s}_{1:T}, \boldsymbol{y}_{1:T}|c)$ depending on the task identity $c \in \{1, \dots, N_c\}$. These sequences are structured according to particular segments (often informally referred to as 'epochs') such as a fixation period, a stimulus presentation period, or a response period. Each of these epochs defines particular distributions of inputs and target responses

presented to the learner. What makes a family of tasks compositionally related is that they are composed from the same underlying set of epochs, albeit with task-dependent transitions from one epoch to the next. Fig. 1a illustrates this schematically for the set of tasks we will consider throughout the paper. We first sought to formalize the notion of shared epochs (Fig. 1b). We can describe complex dependencies in the distribution of inputs and target responses $p(\boldsymbol{s}_{1:T}, \boldsymbol{y}_{1:T}|c)$ for a task by introducing a discrete latent variable $z_t \in \mathbb{Z}^+$ denoting the task epoch at time $t$. We model epochs as evolving over time with task-dependent Markovian transitions $p(z_t = i|z_{t-1} = j, c) = \Lambda_{ij}^c$. Inputs and target responses $\boldsymbol{q}_t \equiv [\boldsymbol{s}_t, \boldsymbol{y}_t] \in \mathbb{R}^{D_q}$ are drawn from an epoch-dependent distribution. Each epoch may involve a number of different input values, reflecting e.g. different experimental conditions like reach direction or stimulus contrast. We therefore introduce dependence on an additional latent variable, the trial variable $x$, which can capture this trial-specific structure, with $\boldsymbol{q}_t \sim p(\boldsymbol{q}_t|z_t, x)$ (Fig 1a). For example, $x$ may index a stimulus, which differs from one trial to the next and determines the distribution of inputs and target responses (see Figure 2c for examples). Importantly, sharing the same $x$ across the entire trial allows the model to capture dependencies between, e.g., the target responses in a later epoch and inputs from an earlier epoch.

Using this description of compositional cognitive tasks, a task-relevant context is naturally defined as the time-varying sequence of task epochs $z_t$. Thus, performing contextual inference requires learning the statistical structure and dependencies of the underlying generative model, and inferring the latent task epoch from observations $\boldsymbol{q}_t$. In the continual learning setting, this is essentially a problem of online learning and inference, which we address in section 3.3. We next give an overview of the specific tasks we consider throughout the rest of the paper. More detailed descriptions of the generative model and the specific task design are provided in Appendix A.1.

## 3.2 Overview of our task set

We initially focus on six commonly studied neuroscience tasks [11, 13, 14, 16] that can be implemented in our generative framework. The tasks are schematized in Fig. 1a and we will henceforth refer to them as `DelayPro`, `DelayAnti`, `MemoryPro`, `MemoryAnti`, `DMPro`, and `DMAnti`. All of these tasks require learning simple stimulus-response relationships, albeit with more complex temporal structure involving e.g. delayed responses. We summarize the different tasks in terms of their general goal and epoch structure in Table 1. In practice, there is substantial inter-trial variability within a task due to variability in the amount of time spent in a given epoch, variability in the trial variable $x$, and additional variability in $\boldsymbol{q}_t = [\boldsymbol{s}_t, \boldsymbol{y}_t]$. All tasks involve a 5-dimensional time series of inputs $\boldsymbol{s}_t$ and a 3-dimensional time series of target responses $\boldsymbol{y}_t$, which take on noisy values around a piecewise constant mean. The value of the mean is dependent on the task epoch $z$ and trial variable $x$, chosen to reflect the underlying task goals. We refer to Appendix A.2 for a more detailed description of the tasks, their epoch structures, and distributional choices.

Note that inclusion of the trial variable $x$ has important consequences for how tasks are segmented into epochs. Considering the `MemoryPro/Anti` tasks: During the *stimulus* epoch ($S$), the inputs reflect the trial variable $x$, which indexes the stimulus direction for each trial. During the 'response period' ($R_{M,P/A}$), the target response is along/opposite the stimulus direction, thus making their dependencies on $x$ different. This necessitates treating them as two different epochs, $R_{M,P}$ and $R_{M,A}$ (see Table 1). While this segmentation arises from our model of the complex statistical dependencies of cognitive tasks, different compositions may be possible using alternative formulations. For instance, one might imagine that all response-epochs could utilize the same context irrespective of how it relates to earlier stimuli, e.g. to move an effector in a particular direction. Indeed, empirical results suggest that this solution arises in RNN models trained on multiple tasks [14].

## 3.3 *What* system: online learning and inference of the compositional task structure

We next consider how the compositional structure across tasks can be learned online (one trial at a time) and continually (one task at a time). The training data contains the sequences of inputs and target responses, as well as the task identity for each trial, $\{(\boldsymbol{q}_{1:T_r}^r, c_r)\}_{r=1,\ldots,N_c N_{trials}}$, where $N_{trials}$ is the number of trials per task. We sometimes drop the trial index $r$ from notations for brevity. Given the training data, the goal of the *what* system is to obtain estimates of the parameters that describe the distribution $p(\boldsymbol{q}_t|c)$ and transitions between epochs. It then infers the unobserved latent variables $z_t$ and $x$ on each trial.

| Task name | Description | Epoch sequence |
|---|---|---|
| DelayPro | Respond towards stimulus direction $\theta$ after a delay. | $F \to S \to R_P$ |
| DelayAnti | Same as DelayPro but respond towards $\theta + \pi$. | $F \to S \to R_A$ |
| MemoryPro | Memorize stimulus direction $\theta$ and respond towards it. | $F \to S \to M \to R_{M,P}$ |
| MemoryAnti | Same as MemoryPro but respond towards $\theta + \pi$. | $F \to S \to M \to R_{M,A}$ |
| DMPro | Compare amplitudes of two stimuli in directions $\theta$ and $\theta'$ and respond towards the direction of the higher amplitude. | $F \to S_{DM} \to R_{DM,P}$ |
| DMAnti | Same as DMPro but respond towards lower amplitude stimulus direction. | $F \to S_{DM} \to R_{DM,A}$ |

Table 1: Our framework captures an ensemble of commonly used cognitive tasks with a shared compositional structure. The epochs are $F$: fixation; $S$: stimulus; $R_{P/A}$: response towards/opposite from the stimulus angle; $M$: memory; $R_{M,P/A}$: response towards/opposite from the memorized stimulus; $S_{DM}$: decision stimuli; $R_{DM,P/A}$: response towards/opposite the stronger stimulus.

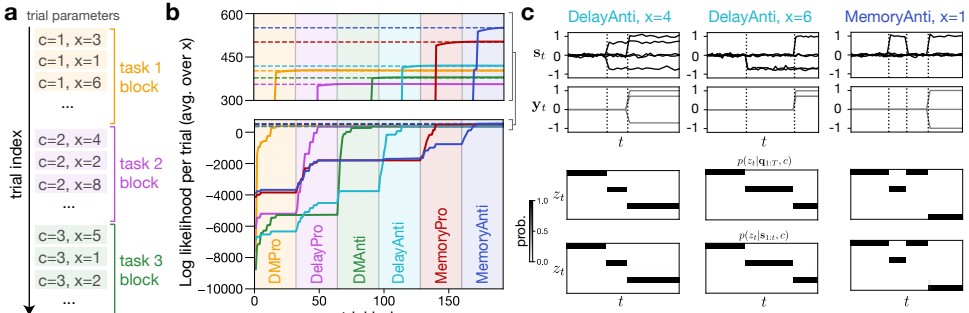

Figure 2: Online continual learning of the compositional structure of tasks. **a**: Schematics of online continual learning of a set of compositional tasks. **b**: Colored lines: log likelihood (LL) of tasks (averaged over trials) from the learned model over the course of learning. Background colors: the task being trained at each trial. Dashed horizontal lines represent LLs computed using ground-truth parameters. **c** Single-trial epoch inference by the learned model. Top row: inputs and target responses of example trials. Dashed vertical lines indicate boundaries between epochs. Trial indices are dropped for brevity. Center row: output of training-time inference ($p(z_t|\boldsymbol{q}_{1:T}, c)$). Bottom row: output of test-time inference ($p(z_t|\boldsymbol{s}_{1:t}, c)$), which does not require access to the target responses or information from the future. Note that, since the $F$ and $M$ epochs in memory-guided tasks have indistinguishable observation models, they are merged by the learner.

When training trials are available as batches across all trials and tasks, inference and learning can be solved using the classic Expectation-Maximization (EM) algorithm. EM performs a coordinate ascent on the complete data log-likelihood via alternating updates to the model parameters and to expected sufficient statistics of the latent variables [37]. To learn online from incoming streams of trials, we maintain estimates of the full-dataset sufficient statistics and update them incrementally after each trial. A pseudocode of the algorithm and details on the inference and parameter update equations are available in Appendix B.1. To overcome convergence to bad local optima in the log-likelihood, we develop a structured incremental initialization procedure described in detail in Appendix B.2.

We applied our algorithm to the problem of online continual learning of the 6 compositional tasks introduced in section 3.2. To evaluate its performance, we computed the average single-trial log likelihood (LL) of different tasks, using the estimated parameters over the course of learning. We found that the model can recover LL on par with that from ground-truth parameters (Fig. 2b), and learning the structure of later tasks does not cause forgetting of previous tasks, even though the observation models of some epochs are shared across tasks. We note that this performance is robust to changes in task ordering (see examples in Appendix B), and the algorithm does not need to know the total number of tasks or epochs in advance, recognizing unfamiliar epochs on the fly.

The learned parameters can be used to infer the epoch identity $z_t$ during each trial, and we considered two types of inference. The more accurate inference (used to produce Fig. 2b) is $p(z_t|\boldsymbol{q}_{1:T}, c)$. This uses both the inputs $\boldsymbol{s}_{1:T}$ and targets $\boldsymbol{y}_{1:T}$ for inference and produces a smoothing distribution in that

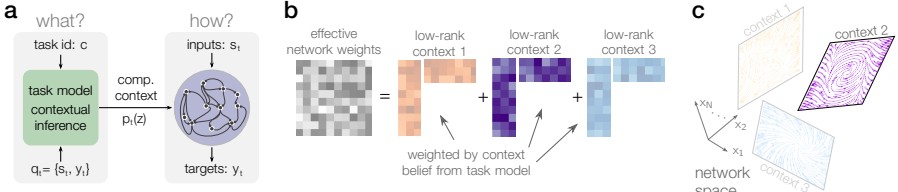

Figure 3: Schematic illustration of two interacting learning systems and contextual gating of RNN dynamics. **a**: The task model (what) processes incoming input and target pairs $\boldsymbol{q}_t = \{\boldsymbol{s}_t, \boldsymbol{y}_t\}$ and infers the time-varying computational context of the current task. A downstream RNN (how) learns the relevant input-to-target transformation and is modulated by the task model's belief $p_t(z)$ over the current context. **b**: The effective recurrent weights of the downstream RNN are a linear combination of low-rank components, weighted by belief over each corresponding context. **c**: The gating by the task model causes the network to express different low-dimensional dynamics across different computational contexts, each of which can be re-expressed and reused to execute different tasks.

the inference of $\boldsymbol{z}_t$ depends on data from $t' > t$. Since this requires access to the target responses, we call this 'training time inference.' (Fig. 2c, center row). We also utilize a 'test time inference' setting, where the learner has only access to the inputs up to the current time point, $\boldsymbol{s}_{1:t}$. This produces a filtering distribution, $p(z_t|\boldsymbol{s}_{1:t}, c)$, which turns out to be highly accurate after learning (Fig. 2c, bottom row). In particular, it is able to disambiguate situations (e.g., $R_{M,P}, R_{M,A}$ at the end of memory-guided response tasks have identical $\boldsymbol{s}_t$ statistics) by using the ground truth task label and the learned epoch transition structure for each task (e.g., knowing that `MemoryPro` and `MemoryAnti` tasks respectively end with $R_{M,P}, R_{M,A}$). Hereafter, we will refer to this module that learns and infers underlying task structures as the 'task model', and to the inferred epochs as 'contexts'.

## 4 Composing recurrent computation across multiple tasks via shared contexts

The results presented so far were concerned with using information available based on the input and response pairs of the task to infer *what* to do. This was achieved through the construction of a probabilistic model capturing the distribution across a family of compositionally related tasks, and the development of an algorithm that can infer a computational context within that task family. We now turn to the problem of using this context to guide the selection and composition of different computations in an RNN that learns *how* to implement the relevant stimulus/response transformations. The general framework for how the two systems interact is illustrated in Fig. 3.

### 4.1 Recurrent neural network architecture

We chose a simple RNN architecture, where the hidden-state activity $\boldsymbol{h}_t$ evolves in time according to

$$\boldsymbol{h}_t = (1 - \alpha)\boldsymbol{h}_{t-1} + \alpha \left[ W^{rec}\phi(\boldsymbol{h}_{t-1}) + W^{in}\boldsymbol{s}_t + \boldsymbol{b}^{in} + \sqrt{2\alpha^{-1}\sigma_r^2}\,\boldsymbol{\xi}_t \right] \tag{1}$$

with decay rate $\alpha$, activation function $\phi$, external input $\boldsymbol{s}_t$, input bias $\boldsymbol{b}^{in}$ and uncorrelated Gaussian noise $\boldsymbol{\xi}_t$. The network activity is read out via

$$\hat{\boldsymbol{y}}_t = W^{out}\phi(\boldsymbol{h}_t) + \boldsymbol{b}^{out}, \tag{2}$$

with output bias $\boldsymbol{b}^{out}$. Given the external input and initial state, the network is trained to produce a target response time series $\boldsymbol{y}_t$ by minimizing the weighted mean squared error between $\boldsymbol{y}_t$ and $\hat{\boldsymbol{y}}_t$ (Appendix C.3).

Inspired by related works on motif execution [16, 38], the connectivity weights in our RNN are dynamically modulated by the posterior belief over computational contexts $p_t(z)$ inferred by the task model (Fig. 3a). More specifically, each computational context in our design corresponds to a set of weights: a low-rank component of recurrent weights $U_z V_z^T$, input weights $W_z^{in}$, input bias $\boldsymbol{b}_z^{in}$, output weights $W_z^{out}$, and output bias $\boldsymbol{b}_z^{out}$. The effective network weights at time $t$ are a weighted

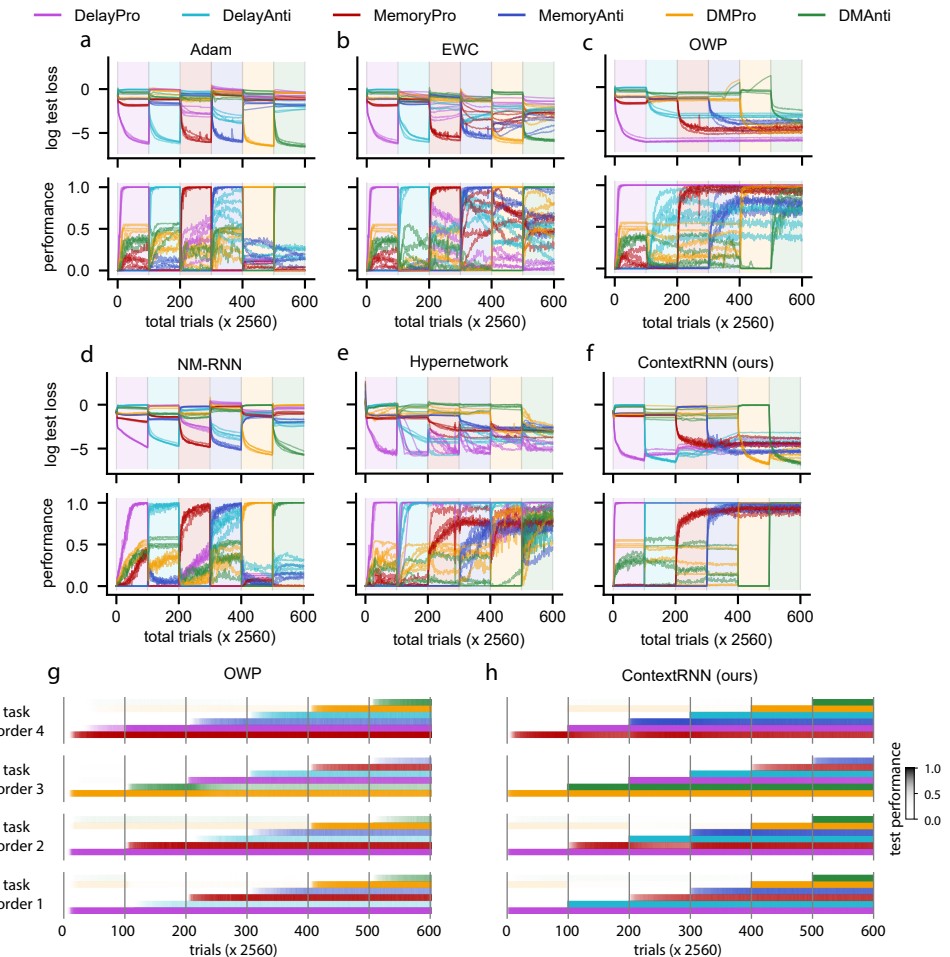

Figure 4: Continual learning. **a-f**: log test loss (colored lines, first row) and test performance (colored lines, second row) throughout sequential training, each plot shows results for five random seeds. Background colors denote the task being trained at each trial. **a**: Adam optimizer on general RNNs. **b**: Elastic Weight Consolidation (EWC) on general RNNs. **c** Orthogonal Weight-Space Projection (OWP) on general RNNs. **d**: Adam optimizer on neuromodulated RNN. **e** Adam optimizer on Hypernetwork RNN. **f**: task model with context modulated RNNs (ours). **g-h**: Color-coded test performance during sequential training of four different task orders. Each row color-codes the average test performance across five random seeds of a specific task over training. We compare OWP (**g**) with our method (**h**).

sum of these components (Fig. 3b):

$$W^{rec} = \sum_z p_t(z)\, U_z V_z^T \tag{3}$$

$$W^{in} = \sum_z p_t(z)\, W_z^{in}, \quad \boldsymbol{b}^{in} = \sum_z p_t(z)\, \boldsymbol{b}_z^{in} \tag{4}$$

$$W^{out} = \sum_z p_t(z)\, W_z^{out}, \quad \boldsymbol{b}^{out} = \sum_z p_t(z)\, \boldsymbol{b}_z^{out} \tag{5}$$

This architecture allows the network to express distinct dynamics for different computational contexts. (illustrated in Fig. 3c). And computations performed by this downstream *how* system are selectively gated by the *what* system. Contrary to previous work, e.g. Duncker et al. [13], we do not constrain the RNN dynamics associated with one context to be non-interfering with those of another.

## 4.2 Contextual modulation enables continual learning

Given the model architecture above, we now turn to the problem of training both systems in parallel on the set of tasks introduced in section 3.2. As before, all training was performed sequentially, where $\{\boldsymbol{s}_t, \boldsymbol{y}_t\}$ pairs from only one task were shown to the network for a number of trials until the task identity switched and the previous tasks were never revisited (illustrated in Fig. 2a). During sequential training of each task, we first ran the incremental EM algorithm (Algorithm 1) for the task model over a whole batch, then computed contextual inference $p_t(z) := p(z_t | \boldsymbol{s}_{1:t}, \boldsymbol{y}_{1:t}, c)$ for the batch to instruct the downstream RNN. During testing, the RNN instead received test time inference $p_t(z) := p(z_t | \boldsymbol{s}_{1:t}, c)$ from the task model, which did not make reference to the target response $\boldsymbol{y}_t$.

We compared the performance of our approach to five baselines and summarized the results in Fig. 4, which shows the log loss and performance (definition see Appendix C.3) on test trials.

There of the baselines operating on general RNNs governed by update equations (1) and (2), where $W^{rec}, W^{in}, \boldsymbol{b}^{in}, W^{out}, \boldsymbol{b}^{out}$ are unconstrained and shared across tasks. The general RNNs additionally receive a task identity input $W^c \boldsymbol{c}$, where $\boldsymbol{c}$ is a one-hot encoding of the current task. Under our default choices of hyperparameters (Appendix C.1), these unconstrained RNNs have approximately twice as many trainable parameters as our context-modulated RNN. The first baseline is to naively train general RNNs using the Adam optimizer [39] without any additional measures for continual learning. In this case, interference across tasks led to catastrophic forgetting, as is readily visible from the log-loss and the task performance, which quickly degraded after a change in task identity (Fig. 4a). The second baseline is Elastic Weight Consolidation (EWC) from Kirkpatrick et al. [31], a continual learning approach which selectively slows down learning rates of single network weights deemed important on previous tasks. Due to the challenging nature of continual learning in RNNs, the simple importance-weighted approach failed to mitigate catastrophic forgetting (Fig. 4b). For a baseline that specifically addressed continual learning in RNNs, we included the approach from Duncker et al. [13], which selectively slows learning rates in subspaces the network explored on previously learned tasks by projecting away these dimensions from the weight update throughout learning. We refer to this method as *Orthogonal Weight-Space Projection* (OWP). While this approach outperformed Adam and EWC, the network suffered from capacity limitation where tasks progressively got harder to optimize. As a result, it became harder for OWP to achieve proficient task performance across all tasks (Fig. 4c).

The other two baselines are more similar to our two-system approach in that they have a dedicated component that receives the task identity input and controls the weights used by an RNN that only receives the task input $\boldsymbol{s}_t$. One of them is neuromodulated RNNs from Costacurta et al. [16], in which a neuromodulatory subnetwork RNN receives task identity input and outputs a signal that dynamically scales the low-rank recurrent weights of an output-generating RNN. Both subnetwork RNNs are jointly optimized end-to-end. While this design is similar to our two-system approach, it did not automatically solve continual learning and quickly forgot previously learned tasks when trained on new tasks (Fig. 4d). The fifth baseline uses hypernetworks for continual learning [23, 32]. A hypernetwork receives distinct learnable embeddings for each task and outputs the task-dependent weights for a target network (an output-generating RNN in our case). Changes to the weights for previously learned tasks are penalized when learning new tasks. This method showed limited improvement over EWC and forgot some earlier tasks at the end of training (Fig. 4e). By contrast, Fig. 4f shows results from our algorithm, which combines the task model with the context-modulated RNN. Selectively modulating the RNN based on the inferred context mitigated catastrophic forgetting and allowed the network to maintain high performance on all previously learned tasks throughout sequential training. This result was robust to different task training orders (Fig. 4g, h).

## 4.3 Contextual inference facilitates forward & backward transfer

We next asked whether, given the compositional structure of the task design, a shared computational context across tasks could facilitate transfer learning. Positive forward transfer refers to the phenomenon where the test loss for a given task decreases more rapidly when it is trained after another task, compared to when it is trained from scratch. We observed positive forward transfer between nearly all pairs of tasks (Fig. 5a and Appendix D). An exception was `MemoryPro/Anti` tasks, which exhibited reliable forward transfer primarily when paired with another `Memory` task. When instead training after `Delay` or `DM` tasks, which share fewer overlapping epochs with `Memory` tasks, the model's test loss on `Memory` occasionally converged to a slightly higher final value after a rapid initial

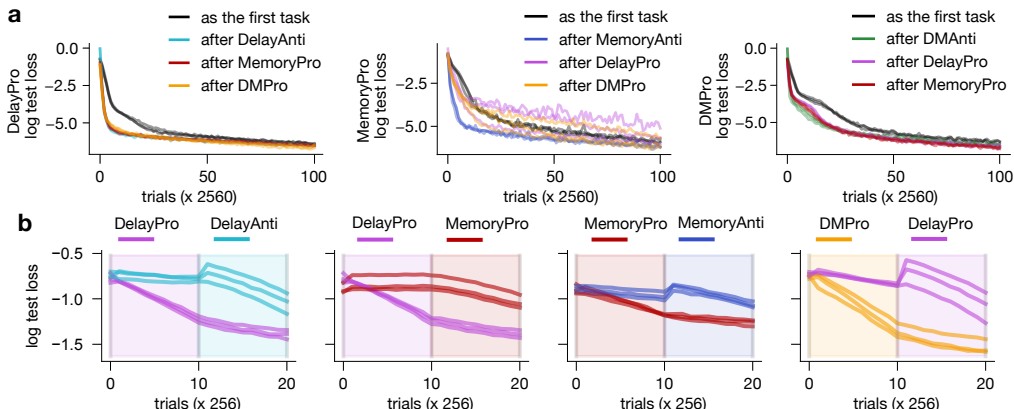

Figure 5: Transfer learning. **a**: The log test loss during training of a task (specified in y-axis label), either as the first task (black) or after training another task (coded by color). We observe faster training after pre-training on tasks that share similar epochs, illustrating forward transfer. **b**: The log test loss during sequential training of two tasks, when each tasks is trained for fewer numbers of trials than our default choice. The loss of the previous tasks continues to decrease after switching to a new task, illustrating backward transfer. We plot results from three random seeds for each task order.

decrease relative to training from scratch. This phenomenon was also observed in Duncker et al. [13]. When each task was trained for fewer trials (switching to a new task before proficiency was reached), we found that its test loss continued to decrease during training on the subsequent task (Fig. 5b and Appendix D), as long as overlapping computational contexts were revisited. This indicates positive backward transfer. Existing continual learning approaches rarely exhibit backward transfer [40]. Indeed, repeating the same experiments using OWP failed to produce improvements on old tasks when learning new ones (see Appendix D).

## 4.4 Contextual selection facilitates compositional generalization on new tasks

As a final experiment, we evaluated the ability of our model to reuse previously learned contexts when encountering new tasks. The ability to do so is called *compositional generalization*, and has been of major interest in neuroscience [14, 15, 36] and machine learning more generally [17–19]. Specifically, we introduced a variant of the memory-guided response task in which the memory epoch ($M$) is omitted. These variants are denoted as M'Pro ($F \rightarrow S \rightarrow R_{M,P}$) and M'Anti ($F \rightarrow S \rightarrow R_{M,A}$) (Figure 6a; note that they are different from DelayPro/DelayAnti tasks in that the stimulus is not shown during response epochs). We initially trained the model sequentially on M'Pro, M'Anti, and MemoryPro ($F \rightarrow S \rightarrow M \rightarrow R_{M,P}$), and then assessed whether it can rapidly learn a new task, MemoryAnti ($F \rightarrow S \rightarrow M \rightarrow R_{M,A}$), within only a few trials. During this final stage, we froze the downstream RNN and updated only the task model, thereby testing whether the model could compose previously learned contexts to solve the novel task. Note that in principle, the freezing of RNN weights was not necessary and that the number of trials needed for learning the task model was generally much smaller than a typical batch size used for training the RNN. Our method reached approximately 83% accuracy within as little as 40 trials on average. In contrast, baselines (also pretrained on the M'Pro/Anti and MemoryPro tasks) using Adam, OWP and Hypernetworks respectively achieved only 56%, 53%, 64% accuracy after 512 trials of full model training (Figure 6b, c). These results demonstrate that our model can rapidly generalize to new tasks by recombining previously learned components.

## 5   Discussion

How does the brain learn to solve tasks through compositional computation? We approached this question by first developing a formalism of compositional tasks that is sufficiently expressive to capture a suite of common neuroscience tasks as compositions from a shared vocabulary of task epochs. We then showed how it is possible to learn and solve these tasks continually using a dual-system approach: First, a *what* system performs online probabilistic inference of the relevant compositional structure of tasks – a time-varying, low-dimensional computational context corresponding to the

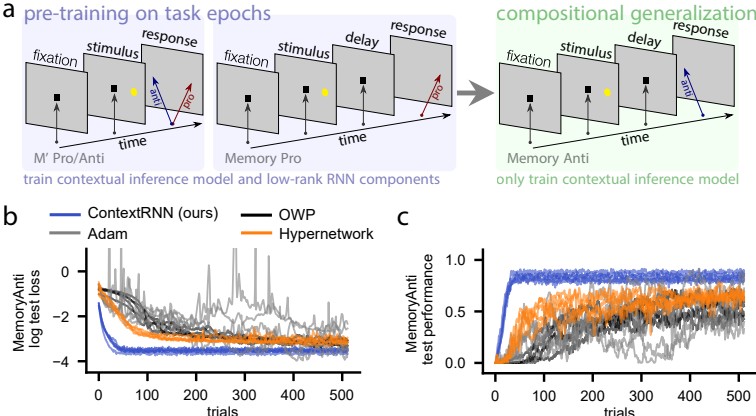

Figure 6: Compositional generalization. **a**: Schematic of pre-training on a set of tasks and learning via contextual inference on a new task. **b-c**: The log loss (**b**) and task performance (**c**) on test trials of the new task as a function of trials used to train the task model (ContextRNN) compared to sequentially training the full models using OWP, Adam and Hypernetwork.

task epoch. This signal is then used to create or (re-)select computational components (one for each context) in a second *how* system. Numerical experiments suggest that incorporating compositionality and contextual inference in this way allows the network to solve the challenging problem of continual learning of compositional tasks. Our approach can utilize knowledge transfer to future and previously encountered tasks, and exhibits potential for compositional generalization.

Studying interactions across different systems and how this may benefit learning, computation, and compositional generalization is also of substantial interest in neuroscience [9, 36, 41] . While we don't directly map our approach to particular candidate brain areas, our work relates to an extensive neuroscience literature in this space. Previous theoretical work has proposed that contextual gating of pattern-generating dynamics for motor sequences can aid flexible recomposition and learning, and be implemented via thalamocortical loops modulated via selective inhibition from basal ganglia [38]. A similar architecture would be possible for our model to extend these ideas from the motor domain to flexible cognitive tasks (see also [27]), and determine how the inhibitory signal from basal ganglia may arise according to contextual information from the *what* system. More generally, there is a large literature on action selection, implicating basal ganglia circuits in the selection and composition of motor sequences [42, 43]. Other lines of work have studied interactions between hippocampus and prefrontal circuits for learning and generalization [3, 44, 45], and how abstract task states in prefrontal- [5, 8] or orbitofrontal cortex [2, 4, 7] may be used to guide goal-directed behavior, computation, and learning.

**Limitations and future work.** A limitation of our work is that we can only infer computational context correctly when this is observable from the observed input and target response pairs of a task. When different epochs map onto the same observations for all $x$ (as is the case in our setting for e.g. fixation and memory epochs), the task model cannot infer that they are different. A possible future extension to overcome this would be to incorporate feedback from the RNN to the task model. A large error produced by the RNN for a familiar context could provide valuable information when different contexts are difficult to distinguish based on external observations alone. Similarly, the segmentation of tasks into epochs is not necessarily unique and depends on our modeling approach. A different notion of shared compositional structure resulting in a different task model may result in a different segmentation and opportunities for reuse. In addition, we have assumed the trial variable $x$ to be a discrete variable, which simplifies the analysis by allowing us to express observation models for different $z, x$ as a look-up table of means. Assuming continuous $x$ may allow the model to capture richer tasks and improve scalability under complex stimulus distributions. Finally, it may be of value to model the inference of unfamiliar epochs in a more principled way by using non-parametric methods (e.g., [1, 20]). A key challenge is to extend existing algorithms to our case where epoch emissions are controlled by a latent factor $x$ that needs to be consistent across epochs, and context variables are shared across multiple tasks.

## Acknowledgments and Disclosure of Funding

We would like to thank Dan O'Shea and Julia Costacurta for feedback on the manuscript and helpful discussions. This work was supported by the National Science Foundation and by DoD OUSD (R&E) under Cooperative Agreement PHY-2229929 (The NSF AI Institute for Artificial and Natural Intelligence), the Gatsby Charitable Foundation (GAT3708), the Kavli Foundation, and the Simons Foundation Collaboration on the Global Brain.

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

# Separating the *what* and *how* of compositional computation to enable reuse and continual learning
# Appendix

All our codes are available on the public repository.

## A Formalization of compositional tasks and task designs

We propose a formalization of shared compositional structures in sequence-to-sequence (seq2seq) tasks and show that the formulation is expressive enough to capture many commonly used tasks in the literature. In each trial of a seq2seq task, the learner receives an input time sequence $\boldsymbol{s}_{t=1,\dots,T}$ and needs to produce a target output sequence $\boldsymbol{y}_{t=1,\dots,T}$. Different tasks (indexed by $c \in \mathbb{Z}^+$) are distinguished from each other by their input distributions as well as the underlying input-to-target rules, together encoded in $p(\boldsymbol{s}_{1:T}, \boldsymbol{y}_{1:T}|c)$. For notational convenience let $\boldsymbol{q}_t \equiv [\boldsymbol{s}_t, \boldsymbol{y}_t] \in \mathbb{R}^{D_q}$.

We model this distribution over task-observables by introducing a set of latent variables resulting in the joint distribution

$$p(\boldsymbol{q}_{1:T}, z_{1:T}, x, c) = p(x|c)p(z_1|c) \prod_{t=1}^{T-1} p(z_{t+1}|z_t, c) \prod_{t=1}^{T} p(\boldsymbol{q}_t|z_t, x) \; p(c). \tag{6}$$

We initially provide intuition about our modeling choices, before formally specifying the distributions for the tasks of the main paper in section A.1 and A.2 below.

To model different tasks with a shared compositional structure, we first observe that in many tasks used in neuroscience experiments, each trial can be temporally segmented into discrete epochs, which we denote as $z_t \in \mathbb{Z}^+$. Each epoch has its distinctive input and target statistics, though they may exhibit complex temporal dependencies across epochs throughout the trial. To illustrate this, we can consider a simple task testing the learner's ability to memorize: During a stimulus epoch, inputs provide information about a trial-specific latent variable (e.g., a dot on the screen indicating an angle $\theta$). The inputs eventually turn off, and the learner needs to maintain the relevant information in memory until a cue solicits a $\theta$-dependent response (e.g., a saccade in the direction of the angle). This example corresponds to the `MemoryPro` task from the main paper.

One might initially attempt to model this epoch dependence as a simple Hidden Markov Model (HMM), where each epoch has its own observation model $p(\boldsymbol{q}_t|z_t)$. However, this approach would fail to capture that the $\boldsymbol{y}_t$ during the response epoch is coupled to the $\boldsymbol{s}_t$ from the stimulus epoch via the shared trial-specific latent variable. In our example above, the statistics of the stimulus epoch and response epoch are coupled since they are both dependent on $\theta$. To take this cross-epoch dependency into account, we explicitly model the "stimulus condition", indexed by the trial variable $x$. In the specific example above, there may be a list of possible directions and $x$ selects one to use as $\theta$. Thus, in our model of cognitive tasks, the observed inputs and target responses depend not only on the latent epoch $z_t$ but also a second latent variable $x$, which additionally parameterizes the observation models of all epochs.

We model tasks as being compositionally related if we can use the same underlying set of task epochs and conditional distributions over $\boldsymbol{q}_t|z_t, x$ to describe them. For a simple example of different tasks sharing compositional structure, we can consider the `MemoryPro` task described above, and construct a second task, `MemoryAnti`, which shares the same stimulus epoch but requires a response in the opposite direction of $\theta$. Here, the dependence of the response epoch on $\theta$ is different (and `MemoryAnti` will therefore have a different response epoch relative to `MemoryPro`), but the statistics of the stimulus epoch are identical across the two tasks. The differences in what epochs contribute to each task and how one epoch transitions to the next is captured via task-dependent Markovian transitions over epochs $p(z_{t+1}|z_t, c)$.

### A.1 Generative model

To make our model of neuroscience tasks tractable and applicable to training RNNs, we make some simplifying choices for its components. For all tasks, $p(x|c)$ is assumed to be a uniform distribution

over some finite set of size $N_x$. $p(z_{1:T}|c)$ is assumed to be a Markov process with the initial state distribution and transition probabilities determined by $c$. Finally, the observation model $p(\boldsymbol{q}_t|z_t, x)$ is assumed to be a multivariate Gaussian with a $(z_t, x)$-dependent mean. Under these assumptions, our model can be seen as an HMM with $c$-dependent latent-state dynamics and $x$-dependent emission models for all latent states. If there is only one task and one possible $x$ value in all trials, the model reduces to a standard HMM. In the more general case, it can be viewed as a mixture (across tasks identities) of HMMs with Gaussian Mixture emissions (across trial variables $x$).

In summary, for trials $r = 1, \ldots, N_{trials}$ we generate tasks as

$$
\begin{aligned}
c_r &\sim p(c) & \text{task identity} && c_r &\in \{1, \ldots, N_c\} \\
z_1^r|c_r &\sim \text{Discrete}(\boldsymbol{\Pi}^c) & \text{epoch identity} && z_t^r &\in \{1, \ldots, N_z\} \\
z_t^r|z_{t-1}^r = z', c_r = c &\sim \text{Discrete}(\boldsymbol{\Lambda}_{:,z'}^c) \\
x_r|c_r &\sim \text{Discrete}(\frac{1}{N_x}\boldsymbol{1}) & \text{trial variable} && x_r &\in \{1, \ldots, N_x\} \\
\boldsymbol{q}_t^r|z_t^r = z, x_r = x &\sim \mathcal{N}\left(\bar{\boldsymbol{q}}_{z,x}, \sigma^2 I\right) & \text{inputs and targets} && \boldsymbol{q}_t^r &\in \mathbb{R}^d
\end{aligned}
$$

This model thus allows us to specify a set of $N_c$ tasks sharing a compositional structure as follows. Let $N_z$ denote the number of epochs shared among these tasks. The composition of the $c$-th task is specified by its epoch-transition parameters: the transition probabilities $\boldsymbol{\Lambda}^c \in \mathbb{R}^{N_z \times N_z}$ : $\Lambda_{z,z'}^c = p(z_t = z|z_{t-1} = z', c)$ and initial probabilities $\boldsymbol{\Pi}^c \in \mathbb{R}^{N_z}$ : $\Pi_z^c = p(z_1 = z|c)$. The observation model of each epoch is specified by the means corresponding to different $x$ values, $\{\bar{\boldsymbol{q}}_{z,x} \in \mathbb{R}^{D_q}\}_{x=1,\ldots,N_x}$, where $\bar{\boldsymbol{q}}_{z,x}$ is the mean of the multivariate Gaussian $p(\boldsymbol{q}_t|z_t = z, x)$. Thus, altogether an ensemble of $N_c$ tasks composed from $N_z$ epochs is specified by the tuple $(\{\bar{\boldsymbol{q}}_{z,x}\}_{z=1,\ldots,N_z;x=1,\ldots,N_x}, \{(\boldsymbol{\Lambda}^c, \boldsymbol{\Pi}^c)\}_{c=1,\ldots,N_c})$, where the first component specifies the epochs and the second specifies how they are used to compose the tasks.

## A.2 Expressing common cognitive and motor tasks in our framework

Below, we provide explicit details of the distributions and parameters used to generate our task set. We follow the same epoch notation introduced in Table 1 of the main paper. We initially describe the temporal structure in terms of epochs of each task, and then define epoch-specific distributions.

All tasks start with the *fixation* epoch ($F$), which presents no stimulus and an active fixation cue. The required output is to maintain fixation without response. In all the non-response epochs (*stimulus* $S$, *memory* $M$ and *decision stimuli* $S_{DM}$), the fixation cue is on and the required output is to maintain fixation without producing a response. During the response epochs ($R_P, R_A, R_{M,P}, R_{M,A}, R_{DM,P}, R_{DM,A}$), the fixation cue turns off and the learner needs to stop fixation and produce a response $\phi$ according to some rule, as described below. Without loss of generality, we considered 8 possible stimulus conditions per task ($N_x = 8$) and generate $x$ i.i.d. from a uniform distribution for the 8 possible values.

**Delayed response tasks** (`DelayPro`, `DelayAnti`). After $F$, the stimulus epoch ($S$) presents an angle $\theta$, chosen from $\{0, \pi/4, ..., 7\pi/4\}$ depending on $x$. The stimulus presentation stays on during the ensuing response epoch ($R_P$ for `DelayPro` or $R_A$ for `DelayAnti`), where the target output becomes $\phi = \theta$ ($R_P$) or $\phi = \theta + \pi$ ($R_A$).

**Memory-guided response tasks** (`MemoryPro`, `MemoryAnti`). After $S$, stimulus presentation disappears in the memory epoch ($M$). During the ensuing response epoch ($R_{M,P}$ for `MemoryPro`, $R_{M,A}$ for `MemoryAnti`), there is still no stimulus presentation and the learner must produce a response based on the memorized $\theta$: $\phi = \theta$ ($R_{M,P}$) or $\phi = \theta + \pi$ ($R_{M,A}$).

**Decision making tasks** (`DMPro`, `DMAnti`). After $F$, a decision stimuli epoch ($S_{DM}$) presents two stimuli simultaneously. $\theta = 0$ with strength $\gamma$ in input dims 1, 2 and $\theta = \pi$ with strength $\gamma'$ in input dims 3, 4. The strengths $(\gamma, \gamma')$ are determined from a set of pairs by $x$ and scale the input channels. During the ensuing response epoch ($R_{DM,P}$ for `DMPro`, $R_{DM,A}$ for `DMAnti`), the stimuli persist and the required output $\phi$ is the direction of the stimulus with a higher strength ($R_{DM,P}$) or the one with a lower strength ($R_{DM,A}$).

The epoch structure of the different tasks is encoded via $\boldsymbol{\Lambda}^c$. Altogether, the 6 tasks can be described as compositions using a shared pool of 10 epochs. Given the epoch identity, the conditional distributions of the inputs and responses are generated as follows:

| epoch | $\bar{\boldsymbol{s}}_{z,x}$ | $\bar{\boldsymbol{y}}_{z,x}$ |
|---|---|---|
| $F, M$ | $[0,0,0,0,0]$ | $[0,0,0]$ |
| $S$ | $[\cos\theta, \sin\theta, 0, 0, 0]$ | $[0,0,0]$ |
| $R_P$ | $[\cos\theta, \sin\theta, 0, 0, 1]$ | $[\cos\theta, \sin\theta, 1]$ |
| $R_A$ | $[\cos\theta, \sin\theta, 0, 0, 1]$ | $[\cos(\theta+\pi), \sin(\theta+\pi), 1]$ |
| $R_{M,P}$ | $[0,0,0,0,1]$ | $[\cos\theta, \sin\theta, 1]$ |
| $R_{M,A}$ | $[0,0,0,0,1]$ | $[\cos(\theta+\pi), \sin(\theta+\pi), 1]$ |
| $S_{DM}$ | $[\gamma\cos\theta, \gamma\sin\theta, \gamma'\cos\theta', \gamma'\sin\theta', 1]$ | $[1,0,0]$ |
| $R_{DM,P}$ | $[\gamma\cos\theta, \gamma\sin\theta, \gamma'\cos\theta', \gamma'\sin\theta', 1]$ | $[\cos\phi, \sin\phi, 1]$, where $\phi = \mathbf{1}_{\gamma>\gamma'}\theta + \mathbf{1}_{\gamma'>\gamma}\theta'$ |
| $R_{DM,A}$ | $[\gamma\cos\theta, \gamma\sin\theta, \gamma'\cos\theta', \gamma'\sin\theta', 1]$ | $[\cos\phi, \sin\phi, 1]$, where $\phi = \mathbf{1}_{\gamma<\gamma'}\theta + \mathbf{1}_{\gamma'<\gamma}\theta'$ |

Where $\mathbf{1}$ is an indicator that takes on value 1 if the subscript is true and 0 otherwise. This fully specifies the conditional distributions $p(\boldsymbol{q}_t|z_t, x)$ for each task epoch with $\bar{\boldsymbol{q}}_{z,x} = [\bar{\boldsymbol{s}}_{z,x}, \bar{\boldsymbol{y}}_{z,x}]$. For the DM tasks, we restrict the stimulus values to two locations $\theta = 0, \theta' = \pi$, but pick $N_x = 8$ different combinations of possible $(\gamma, \gamma')$ pairs. Note that while we generate the input and response distributions using stimulus values $\theta$ to follow convention from the literature [11, 14], each value of $\theta$ (or $(\gamma, \gamma')$ in the DM tasks) maps onto a different $x$ value. For $\sigma$ we used 0.05; for $(\gamma, \gamma')$, $x$ selects from $[0.5, 1], [1, 2], [0.5, 2], [0.2, 1.5], [1, 0.5], [2, 1], [2, 0.5], [1.5, 0.2]$.

Note that while we have followed many conventions from previous work in the task design, these previous approaches tend to implement each task individually but with related distributional assumptions [11, 13, 14, 16]. Modeling shared structure across tasks through an explicit, shared generative framework for an entire task family is novel. While this was not the focus of the main paper, it is worth noting that access to a description of shared statistical structure in the input and target response pairs of each task forms an important baseline for expectations of shared statistical structure across tasks in the solution emerging after RNN training [11, 13–15].

# B  Online learning and inference of compositional task structures

## B.1  Online learning and inference

In this section, we provide additional details on the algorithms developed for performing posterior inference over the latent variables of the task model, and online (one-trial-at-a-time) learning of the task model.

**Inference.**  We can perform exact inference by utilizing a message passing scheme similar to that used for performing inference in classic HMM models.

Let $\alpha_t^r(z, x, c) = p(\boldsymbol{q}_{1:t}^r, z_t^r = z | x_r = x, c_r = c)$ denote the forward (filtering) message for a given task. During a filtering pass, we compute

$$\alpha_1^r(z, x, c) = p(z_1^r = z | c_r = c) p(\boldsymbol{q}_1^r | z_1^r = z, x_r = x) \tag{7}$$

$$\alpha_{t+1}^r(z, x, c) = \left( \sum_{z'=1}^{N_z} \alpha_t^r(z', x, c) p(z_{t+1}^r = z | z_t^r = z', c_r = c) \right) p(\boldsymbol{q}_{t+1}^r | z_{t+1}^r = z, x_r = x) \tag{8}$$

Marginalizing the forward message at the final time-step allows us to compute the marginal likelihood over observations

$$p(\boldsymbol{q}_{1:T}^r | x_r = x, c_r = c) = \sum_{z=1}^{N_z} \alpha_T^r(z, x, c) \tag{9}$$

$$\tag{10}$$

Let $\beta_t^r(z, x, c) = p(\boldsymbol{q}_{(t+1):T}^r | z_t^r = z, x_r = x, c_r = c)$ denote the backwards (smoothing) message. During the smoothing pass, we compute

$$\beta_T^r(z, x, c) = 1 \tag{11}$$

$$\beta_t^r(z, x, c) = \sum_{z'=1}^{N_z} p(z_{t+1}^r = z' | z_t^r = z, c_r = c) p(\boldsymbol{q}_{t+1}^r | z_{t+1}^r = z', x_r = x) \beta_{t+1}^r(z', x, c) \tag{12}$$

Given these quantifies, we can compute the joint posterior over $z_t^r$, $x_r$ and $c_r$ as

$$\gamma_t^r(z, x, c) = p(z_t^r = z, x_r = x, c_r = c | \boldsymbol{q}_{1:T}^r) \tag{13}$$

$$= \frac{p(\boldsymbol{q}_{1:t}^r, z_t^r = z | x_r = x, c_r = c) p(\boldsymbol{q}_{(t+1):T}^r | z_t^r = z, x_r = x, c_r = c) p(x_r = x, c_r = c)}{\sum_{x,c} p(\boldsymbol{q}_{1:T}^r | x_r = x, c_r = c) p(x_r = x, c_r = c)} \tag{14}$$

$$= \frac{\alpha_t^r(z, x, c) \beta_t^r(z, x, c) p(x_r = x, c_r = c)}{\sum_{x,c} \sum_{z'=1}^{N_z} \alpha_T^r(z', x, c) p(x_r = x, c_r = c)} \tag{15}$$

Note that the filtering and smoothing passes can be done for each value of $x$ and $c$ in parallel. Finally, for later use in the online learning algorithm we also compute the joint posterior over $z_{t-1}^r$, $z_t^r$, $x_r$ and $c_r$ as

$$\xi_t^r(z, z', x, c) = p(z_{t-1}^r = z, z_t^r = z', x_r = x, c_r = c | \boldsymbol{q}_{1:T}^r) \tag{16}$$

$$= \frac{\alpha_{t-1}^r(z, x, c) p(z_t^r = z' | z_{t-1}^r = z, c_r = c) p(\boldsymbol{q}_t^r | z_t^r = z', x_r = x) \beta_t^r(z', x, c) p(x_r = x, c_r = c)}{\sum_{x,c} p(\boldsymbol{q}_{1:T}^r | x_r = x, c_r = c) p(x_r = x, c_r = c)} \tag{17}$$

**Online learning.** Learning aims to recover parameters of the generative model, $(\{\bar{\boldsymbol{q}}_{z,x}\}_{z=1,\dots,N_z; x=1,\dots,N_x}, \{(\boldsymbol{\Lambda}^c, \boldsymbol{\Pi}^c)\}_{c=1,\dots,N_c})$, up to a permutation over $z$ and $x$. Performing parameter learning with EM requires computing the expected counts of visiting particular epochs or transitioning across epochs across all trials. When all trials are available as a batch, this involves sums over the quantities computed during inference for each trial. To make notation more compact, we denote $X$ as the set of sufficient statistics needed to update the set of model parameters $\Theta$. Let $\Theta^{(i,k)}$ denote the learned parameters after seeing the $i$-th trial and running $k$ EM iterations. Let $X^{r,(i,k)}$ denote the single-trial statistics on trial $r$, computed using parameters $\Theta^{(i,k)}$. When the learning algorithm has access to all trials throughout learning (batch EM), the updates take the form

$$\Theta^{(i,k+1)} = f(S^{(i,k)}(X)) \quad S^{(i,k)}(X) = \sum_{r=1}^i X^{r,(i,k)}. \tag{18}$$

For example, for the parameter estimates for the transition matrix, this takes the form

$$\Lambda_{czz'}^{i,k+1} = \frac{\sum_{t=1}^T \sum_{x=1}^{N_x} \Xi_{cxzz'}^{i,k}(t)}{\sum_{t=1}^T \sum_{x=1}^{N_x} \sum_{z''=1}^{N_z} \Xi_{cxz''z'}^{i,k}(t)}. \tag{19}$$

with

$$\Xi_{cxzz'}^{i,k}(t) = \sum_{r=1}^i p\left(c_r = c, x_r = x, z_{t-1}^r = z, z_t^r = z' | \boldsymbol{q}_{1:T}^r; \Theta^k\right). \tag{20}$$

In general, the batch EM parameter updates take well-known forms for HMMs and GMM-HMMs, which is why we only give an example here in the interest of brevity.

When trials are only available one-trial-at-a-time and cannot be revisited (online EM), the sums of expected sufficient statistics across trials have to be approximated and updated after each trial instead, leading to an approximation to the batch updates. We propose the following update rule to perform parameter learning online

$$\Theta^{(i,k+1)} = (1 - \eta_{params}) \Theta^{(i-1,K)} + \eta_{params} f(S_{online}^{(i,k)}(X)). \tag{21}$$

$$S_{online}^{(i,k+1)}(X) = (1 - \eta_{stats}) S_{online}^{(i-1,K)}(X) + X^{i,(i,k)}. \tag{22}$$

where $K$ denotes the number of iterations per trial. After learning the $i$-th trial, we only need to store $\Theta^{(i,K)}$ and $S_{online}^{(i,K)}(X)$, taking the form

$$\Theta^{(i,K)} = \sum_r^i (1 - \eta_{params})^{i-r} \eta_{params} f(S_{online}^{(r,K)}(X)) + (1 - \eta_{params})^{i+1} \Theta_{init.} \tag{23}$$

$$S_{online}^{(i,K)}(X) = \sum_r^i (1 - \eta_{stats})^{i-r} X^{r,(r,K)}. \tag{24}$$

We summarize this online learning approach in Algorithm 1 and show performance for different training orders (supplementing Figure 2 in the main paper) in Figure S.1.

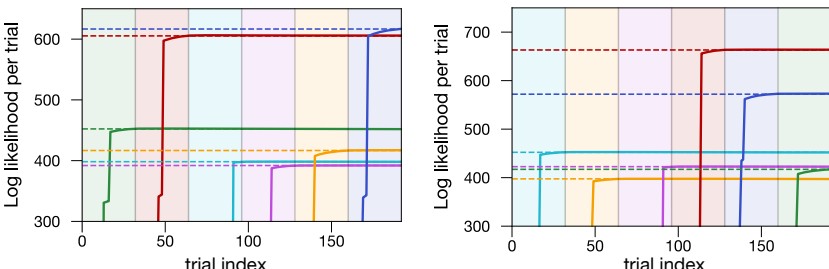

Figure S.1: Online continual learning of task structures with some example task orderings. See Fig.2 for legends.

---

**Algorithm 1** High-level Overview of the Incremental EM algorithm.

---

$\Theta \leftarrow$ init.            $\triangleright$ Initialize parameter estimates.
$S(X) \leftarrow \mathbf{0}$      $\triangleright$ These are the estimated sums of sufficient statistics across trials
**for** $r = 1, ..., N_{trials}$ **do**
     $\Theta \leftarrow incremental\_initialize(\Theta, q^r_{1:T_r}, c_r)$      $\triangleright$ See B.2.
     $\hat{\Theta} \leftarrow \Theta$      $\triangleright$ Create a temporary copy of the parameters
     $\hat{S}(X) \leftarrow$ None
     **for** $k = 1, ..., K$ **do**
         $X \leftarrow getstats(\boldsymbol{q}^r_{1:T_r}, c_r, \hat{\Theta})$      $\triangleright$ Compute sufficient statistics of this trial with current params.
         $\hat{S}(X) \leftarrow (1 - \eta_{stats})S(X) + X$      $\triangleright$ Decay the sums from the previous trial and add new stats
         $\hat{\Theta} \leftarrow \Theta \odot (1 - \eta_{params}M_{gate}) + \eta_{params}f(\hat{S}(X)) \odot M_{gate}$      $\triangleright$ Incremental update of params. using the modified sums. $M_{gate}$ is a binary mask controlling which parameters are updated (see B.3).
     **end for**
     $\Theta \leftarrow \hat{\Theta}$
     $S(X) \leftarrow \hat{S}(X)$
**end for**

---

## B.2   Incremental initialization of model parameters

Even for simple models such as the Gaussian Mixture Model, EM is susceptible to local optima [46]. To avoid convergence to bad local optima, it is important to obtain good initializations for the model parameters. For simple time-series models such as the HMM, this is typically achieved by collapsing the sequences across time, performing clustering (e.g. with K-means [47]), and using the resulting cluster centers as the initial observation-model parameters, given each latent discrete state.

In our model, we need to initialize the underlying cluster centers of the observation model, $\{\hat{\boldsymbol{q}}_{z,x}\}_{z=1,...,\hat{N}_z, x=1,...,N_x}$. Here, $\hat{N}_z$ reflects the fact that the total number of epochs in the entire dataset is unknown *a priori*. We overcome this by setting up a large number of 'slots' ($\hat{N}_z \geq N_z$), exceeding the likely total number of epochs in the task family. Our setting adds two significant challenges relative to simple HMMs: First, each trial contains only a few epochs and a specific stimulus condition ($x$ value). Therefore, clustering each trial can only initialize a small subset of the $N_x N_z$ mixture components of the observation model. Second, since there is one stimulus condition $x$ per trial, the mixture components $\bar{\boldsymbol{q}}_{z,x}$ explaining a given trial must be allocated to the same $x$, but different epoch states $z$. Initialization schemes based on simple clustering approaches on the entire trial (such as that outlined for HMMs above) would be agnostic to this structure and fail to provide a feasible initial set of parameters. To overcome these issues, we introduce an 'incremental initialization' scheme, which is applied to all the estimated means $\{\hat{\boldsymbol{q}}_{z,x}\}$ before learning each trial.

To introduce the scheme, we first introduce the notion of the putative $z, x$, denoted $\tilde{z}, \tilde{x}$. This is to highlight that learning the correct $\{\bar{q}_{z,x}\}$ does not require inferring the real $z, x$ in the generative process but only requires them to be correct up to a permutation. Let $\tilde{N}_z, \tilde{N}_c$ denote the number of

epochs and task slots in the learning algorithm, respectively. Note that these do not need to be set as the correct $N_z, N_c$ – they can simply be large integers. We do assume that the system knows the correct number of $x$ values, $N_x$. The learner keeps track of $\{\hat{\boldsymbol{q}}_{\tilde{z},\tilde{x}}\}_{\tilde{z}=1,...,\tilde{N}_z,\tilde{x}=1,...,\tilde{N}_x}$ as well as two Boolean-valued tables, $F_{c,x}$ of size $\tilde{N}_c \times N_x$ and $F_{x,z}$ of size $\tilde{N}_x \times \tilde{N}_z$. The two tables keep track of which combinations of $c, \tilde{x}$ and $\tilde{x}, \tilde{z}$ have been encountered.

When the $r$th trial is observed and the learner has access to $(\boldsymbol{q}^r_{1:T_r}, c_r)$, the scheme is threefold:

1. We perform K-means clustering on the entire sequence $\boldsymbol{q}^r_{1:T_r}$. This gives us a set of cluster means. Since we assume the epochs to have piecewise constant inputs and mild noise, the means correspond to the different epochs that appeared in this trial. The challenge now is to assign $\tilde{z}, \tilde{x}$ to these means, and to use them to initialize $\hat{\boldsymbol{q}}_{\tilde{z},\tilde{x}}$ accordingly.

2. All cluster means from step (1) should be assigned the same $\tilde{x}_r$. We check the means against $\{\hat{\boldsymbol{q}}_{\tilde{z},\tilde{x}}\}$ and decide on $\tilde{x}_r$ according to a set of rules and $F_{c,x}$. The $c_r, \tilde{x}_r$ pair is marked 'encountered' in $F_{c,x}$.

3. Given $\tilde{x}_r$ from step (2), we treat the cluster means not found in $\{\hat{\boldsymbol{q}}_{\tilde{z},\tilde{x}}\}_{\tilde{x}=\tilde{x}_r}$ as unfamiliar $\tilde{z}, \tilde{x}$, meaning that they represent previously unseen $\tilde{z}, \tilde{x}$ combinations and should be used to initialize. Each center is assigned a different $\tilde{z}$ that has not been encountered (according to $F_{x,z}$). The $\tilde{z}, \tilde{x}_r$ pairs are marked as 'encountered' in $F_{x,z}$.

## B.3   Gated updates to parameters

Since each trial contains information about only one task and the few epochs it is associated with, it does not make sense to update parameters related to other tasks and epochs. For $\boldsymbol{\Lambda}^c$, we simply gate it such that only the transition matrix corresponding to the current task (the label of which is given) is updated. For $\{\hat{\boldsymbol{q}}_{z,x}\}$, we infer which epochs appeared in this trial using the posterior $p(z^r_{1:T}|\boldsymbol{q}^r_{1:T})$. Only epochs with a sufficiently high chance of appearance have their $\{\hat{\boldsymbol{q}}_{z,x}\}$ updated.

## B.4   Epoch identifiability in our set of tasks

For the particular set of tasks we considered, since $F, M$ epochs have identical $\bar{\boldsymbol{q}}_{z,x}$ for all $x$, they are indistinguishable. Thus, our learning algorithm will combine them into a single epoch, which we denote as $F/M$. Instead of learning the 10 epochs in the generative process for our set of tasks, the task-model will end up learning 9 epochs, including the combined $F/M$ epoch. In terms of the transition matrices, this creates a complication for MemoryPro, MemoryAnti tasks where both $F$ and $M$ epochs appear in each trial. The learned transitions will not be deterministic in the sense that the $F/M$ epoch may transition to either the stimulus epoch or the response epoch. The 'ground-truth' parameters used for plotting in Fig. 2 and Fig. S.1 refer to the optimal parameters with a merged $F/M$ epoch. In future work, it will be interesting to investigate how epochs that are computationally distinct (e.g. holding still in $F$ vs. holding still while maintaining a memory in $M$) but map onto the same observations may be distinguished, e.g. via feedback from the downstream network implementing the different computations for each epoch.

# C  RNN architecture and hyperparameter settings

## C.1  Default RNN architecture and task parameters

| Parameter | Value |
|:---:|:---:|
| $\alpha$ | 0.1 |
| $\sigma_r$ | 0.05 |
| $\phi$ | ReLU |
| number of hidden units | 256 |
| rank of $U_z, V_z$ | 3 |
| input noise std | $\sqrt{2/\alpha}\,\sigma_{in}$, where $\sigma_{in} = 0.01$ |
| minimum duration of a epoch | 5 time steps |
| $p(z_{t+1} = z_t \vert c)$ | 0.9 |

## C.2  Default training protocol

We used a batch size of 256 and trained each task for 1000 batches unless otherwise specified.

For the context-modulated RNN, the learning rate for the weights associated with each context $z$ was initially set to $\eta_z = 0.001$. After training each task $c$, $\eta_z$ was multiplied by a decay factor $\gamma = 0.5$ for any context with $p(z \mid c) > 0.001$. During training on task $c$, $L_2$ weight regularization was applied with a coefficient of $10^{-5}$ for contexts with $p(z \mid c) > 0.001$, and set to 0 for all other contexts.

For baseline algorithms using general ("vanilla") RNN architectures, the learning rate was set to 0.01, and the $L_2$ weight regularization coefficient was set to $10^{-5}$. Parameter choices were determined by a coarse grid search.

## C.3  Loss function and performance measure

The loss function is a weighted mean square error similar to [11, 14]. $L := \langle m_{i,t}(y_{i,t} - \hat{y}_{i,t})^2 \rangle_{i,t}$, where $i$ is the index of the output units, $m_{i,t} = 1$ for response epochs and $m_{i,t} = 0.2$ for all other epochs.

A trial is considered correct if the network maintained fixation for all time steps before the fixation cue turns off, and responded to the correct direction for time steps in the response epoch. If the activity of the fixation output exceeds 0.5, the network is considered to have broken fixation. The network's response direction is considered correct if its angular difference from the target direction is less than $\pi/10$. Average performance and test loss were calculated on 200 test trials for each task.

# D  Supplementary results

## D.1  Additional results on transfer learning

We provide additional results on transfer learning. Figure S.2 and Figure S.3 supplement Figure 5**a** and **b**, respectively, by showing results for all task pairs. Figure S.4 shows the lack of backward transfer when training with the OWP algorithm [13], where the test loss of previously learned tasks did not decrease when training on subsequent tasks with overlapping epochs.

## D.2  Results with other hyperparameter choices

We observed improved performance of our continual learning algorithm as we increased the rank of $U_z$ and $V_z$ (denoted by $r$) (Figure S.5**a,b**, compared with Figure 4**f**). Using $r = 3$, 5, and 10 in ContextRNN results in 34587, 43803, and 66843 trainable parameters, respectively — all fewer than the 69379 parameters of a general RNN with the same number of hidden units. With $r = 3$, ContextRNN performed worse with tanh than with ReLU activation, but this gap was closed at $r = 10$

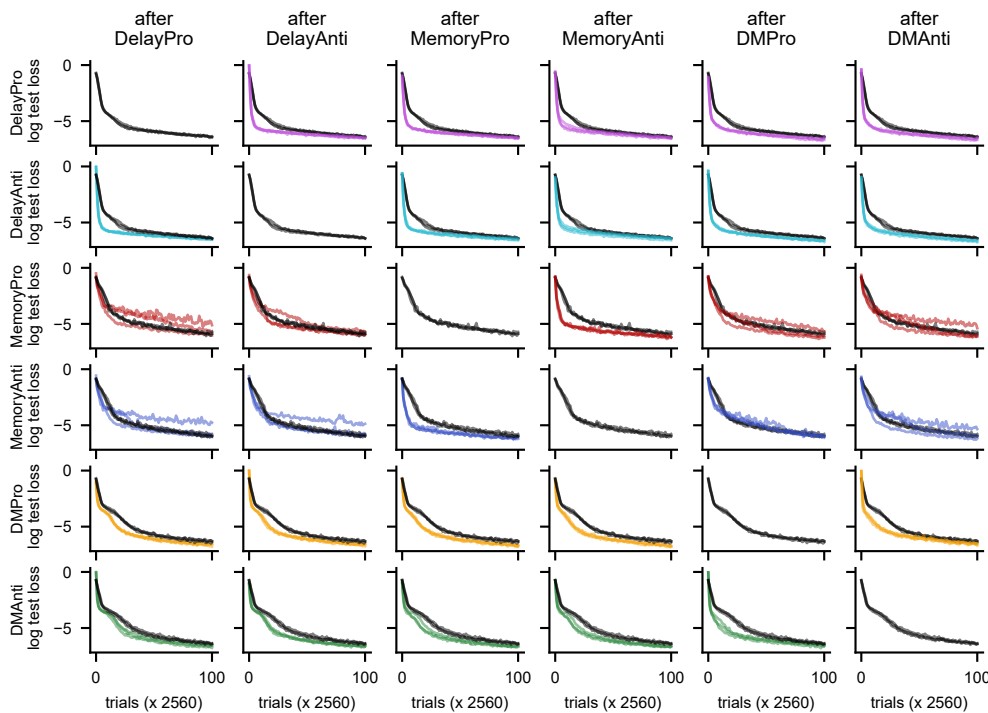

Figure S.2: Additional results on forward transfer. Colored curves show the log test loss of task B (indicated by the row label) when trained after task A (column label). For comparison, black curves in each row show the log test loss of task B when trained from scratch. We plot results across three random seeds for each task order.

(Figure S.5**c**, **d**). Similarly, performance with $r = 3$ and $\alpha = 0.2$ was worse than with $\alpha = 0.1$, but this gap was also closed at $r = 10$ (Figure S.5**e**, **f**).

## E  Details of baseline implementations

### E.1  Elastic Weight Consolidation

Elastic Weight Consolidation (EWC) [31], slows down learning rates for network weights deemed important for previous tasks. This is achieved by adding a regularization term to the training loss function. For a loss $\mathcal{L}(\Theta)$, the EWC objective is given as

$$\mathcal{L}^{EWC}(\Theta) = \mathcal{L}(\Theta) + \frac{\lambda}{2} \sum_i F_i (\Theta_i - \Theta_i^*)^2 \tag{25}$$

Here $F_i$ is an importance weight that ties the $i$th parameter value $\Theta_i$ to it's value $\Theta_i^*$ at the end of training on the previous task. The important weights are computed as the diagonal of the Fisher Information matrix $F$ evaluated at the parameter values $\Theta_i^*$. We set $\lambda$ to $10^5$ after a coarse grid search.

### E.2  Orthogonal Weight Projection

The original learning rule in Duncker et al. [13] was derived for a parameterization of network dynamics of the form

$$\dot{\boldsymbol{x}} = -\boldsymbol{x} + \phi \left( \boldsymbol{W}^{\text{rec}} \boldsymbol{x} + \boldsymbol{W}^{\text{in}} \boldsymbol{s} \right) \tag{26}$$

given an element-wise activation function $\phi(\cdot)$. The learning rule was intended to maintain the stimulus/response relationship of previous task by applying a set of projection matrices to the gradient used to update the network weights over learning. The projection matrices were intended to remove

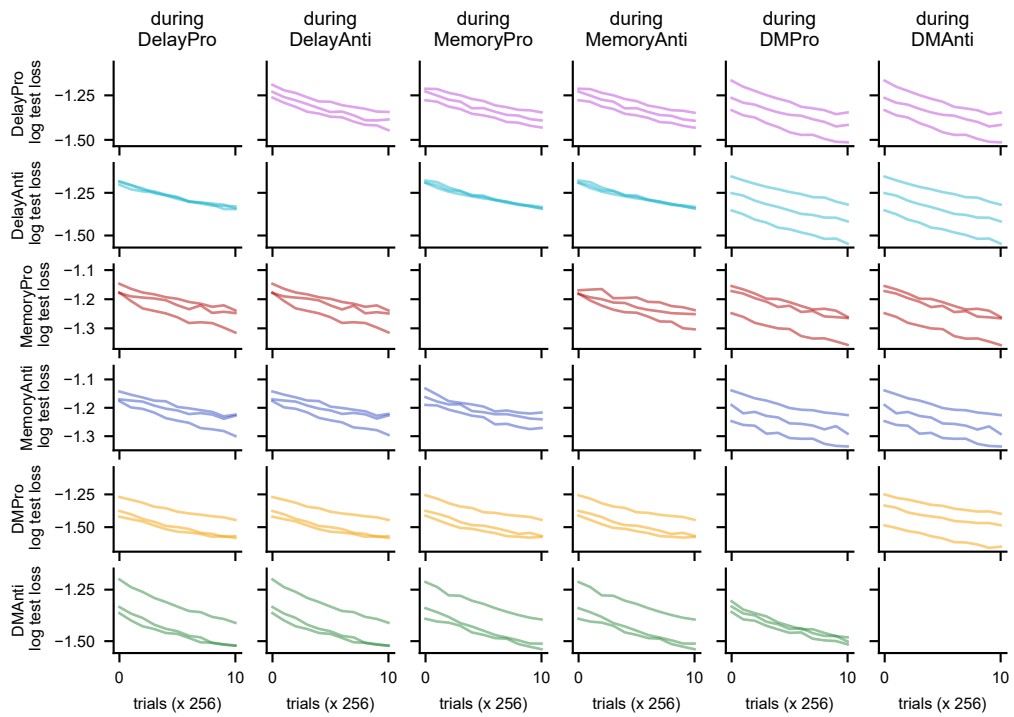

Figure S.3: Additional results on backward transfer. The log test loss of the previously trained task A (indicated by the row label) continue to decrease during subsequent training on another task B (column label). Each task is trained with 2560 trials and we plot results across three random seeds for each task order.

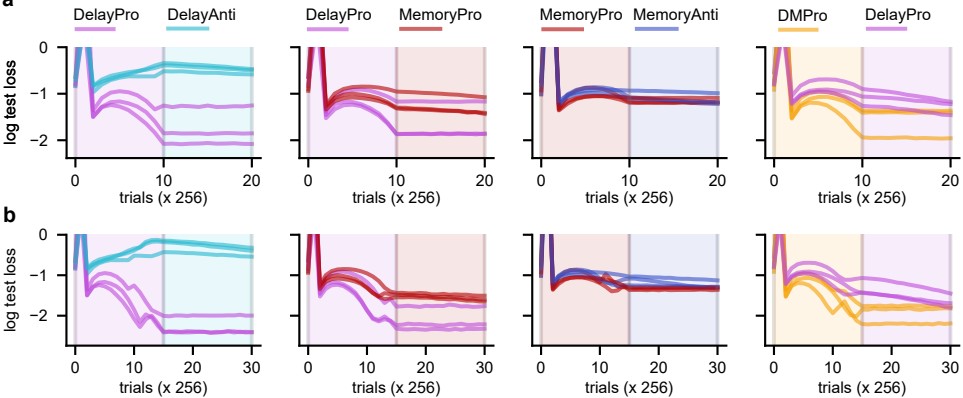

Figure S.4: No backward transfer with OWP. **a**: The log test loss during sequential training of two tasks with the OWP algorithm, each tasks is trained for 2560 trials. The loss of the previous tasks does not decrease after switching to a new task, indicating no backward transfer. Results are shown for three random seeds per task order. **b**: same as **a** but with each task trained for 3840 trials.

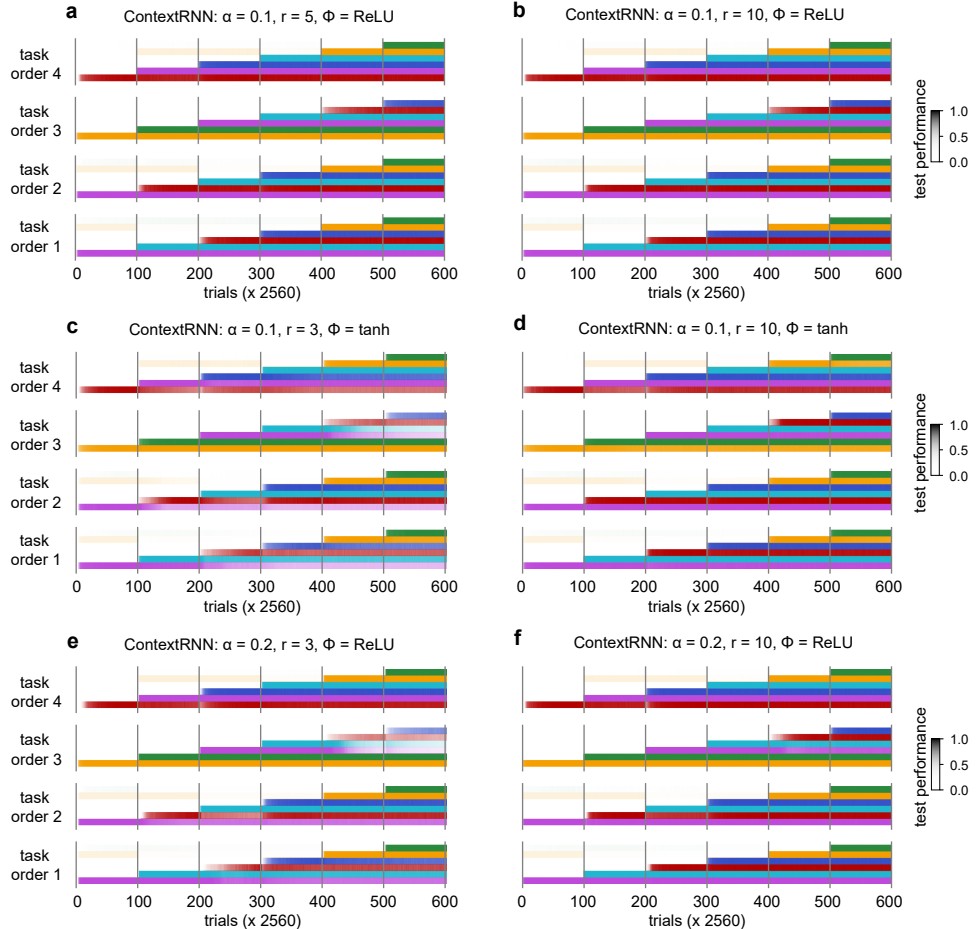

Figure S.5: Continual learning performance with different hyperparameter choices. **a**: Color-coded test performance during sequential training of four different task orders. Each row color-codes the average test performance across five random seeds of a specific task over training. RNNs used for this panel have $\alpha = 0.1, r = 5, \phi = \text{ReLU}$. **b-f**: same as **a** but with a different hyperparameter choice labeled by the title.

directions from the weight update that would interfere with previous tasks and were defined as follows. Letting $\boldsymbol{z}_t^{c,r} = \begin{bmatrix} \boldsymbol{x}_t^{c,r} \\ \boldsymbol{s}_t^{c,r} \end{bmatrix}$ denote the concatenated network state and input state at time $t$ of trial $r$ on task $c$, $\boldsymbol{Z}_{1:c} = [\boldsymbol{z}_1^{1,1}, \ldots \boldsymbol{z}_T^{c,r}]$ the collection of all time points and trials on tasks 1 through $c$, and $\boldsymbol{W} = [\boldsymbol{W}^{\text{rec}} \; \boldsymbol{W}^{\text{in}}]$ the concatenated weight matrices, we define the projection matrices

$$\boldsymbol{P}_1^{1:c} = \left( \frac{1}{\lambda} \boldsymbol{Z}_{1:c} \boldsymbol{Z}_{1:c}^{\mathsf{T}} + I \right)^{-1} \tag{27}$$

$$\boldsymbol{P}_2^{1:c} = \left( \frac{1}{\lambda} \boldsymbol{W} \boldsymbol{Z}_{1:c} \boldsymbol{Z}_{1:c}^{\mathsf{T}} \boldsymbol{W}^{\mathsf{T}} + I \right)^{-1} \tag{28}$$

and the modified learning update as

$$\Delta W^{\text{CL}} \propto \boldsymbol{P}_2^{1:c} \, \nabla_{\boldsymbol{W}} \mathcal{L} \, \boldsymbol{P}_1^{1:c} \tag{29}$$

where $\nabla_{\boldsymbol{W}} \mathcal{L}$ is the derivative of the loss on the new task with respect to the network weights. An analogous set of projection matrices and modified learning update is used for the readout weights.

To facilitate direct comparisons with our approach, we adapted the learning rule to a modified setting, where the RNN dynamics are expressed as

$$\dot{\boldsymbol{h}} = -\boldsymbol{h} + \boldsymbol{W}^{\text{rec}} \phi(\boldsymbol{h}) + \boldsymbol{W}^{\text{in}} \boldsymbol{s} \tag{30}$$

given the same element-wise activation function $\phi(\cdot)$. While the two parameterizations in (26) and (30) are generally considered equivalent [48], the linear intuition used to motivate the approach of [13] should be exact in (30). With $\boldsymbol{x}_t = \phi(\boldsymbol{h}_t)$, $\boldsymbol{Z}_{1:c}$ is unchanged, but we instead use

$$\boldsymbol{P}_2^{1:c} = \left( \frac{1}{\lambda} \boldsymbol{H}_{1:c} \boldsymbol{H}_{1:c}^{\mathsf{T}} + I \right)^{-1} \tag{31}$$

where $\boldsymbol{H}_{1:c} = [\boldsymbol{h}_1^{1,1}, \ldots \boldsymbol{h}_T^{c,r}]$. While this is very similar to the version in Duncker et al. [13], the projections matrix now only depends implicitly on the $\boldsymbol{W}$ of previous tasks. We performed all comparisons using this modified learning rule.

## E.3 Neuromodulated RNN

Costacurta et al. [16] designed a neuromodulated RNN (NM-RNN), in which a neuromodulatory subnetwork (RNN1) outputs a low-dimensional signal that dynamically scales the low-rank recurrent weights of an output-generating RNN (RNN2). RNN1 and RNN2 are jointly optimized through gradient descent. In our implementation, RNN2 has 256 neurons and with rank($\mathbf{W}^{rec}$) = 27, equal to the default choice for our ContextRNN. Number of neurons in RNN1 is set to 125 such that the whole network has 36658 parameters, comparable to the 34587 parameters of our contextRNN. $\alpha$ is set to 0.1 for RNN2 and 0.01 for RNN1 following the original paper. We used Adam optimizer, with learning rate set to 0.001 and $L_2$ weight regularization coefficient set to $10^{-5}$ after a coarse grid search.

## E.4 Hypernetwork

Von Oswald et al. [23] designed task-conditioned hypernetworks to tackle continual learning. A hypernetwork receives distinct learnable embeddings for each task, and outputs the weights for the target network (an output-generating RNN in our case). For the output-generating RNN, we set the number of neurons to 256. A full-rank recurrent weight matrix, an input weight matrix and an output weight matrix together require a 67843-d output from the hypernetwork. We used a chunked hypernetwork introduced in the original paper, where a set of learnable chunk embeddings serve as additional input to the hypernetwork. Concatenating the hypernetwork output from distinct chunks gives all parameters for the target network. We set the output dimension of the hypernetwork to 2000 and number of chunks to 34. Dimension of task embeddings and chunk embeddings is set to 32. The hypernetwork is a multi-layer perceptron with two 32-d hidden layers. Together, the number of trainable parameters of the hypernetwork equals 70416, comparable to other baselines in our paper using a full-rank RNN. After a coarse grid search, we set continual learning regularization strength to 1, orthogonal regularization strength to 0.001 and clip gradient norm to 1 (see Ehret et al. [32] for details). We used Adam optimizer, with learning rate set to 0.001 and $L_2$ weight regularization coefficient set to $10^{-5}$.

