# OpenReview forum: "Separating the 'what' and 'how' of compositional computation to enable reuse and continual learning"
_NeurIPS.cc/2025/Conference — NeurIPS 2025 poster_

### Official Review · Reviewer_yJUA · 2025-06-06

**Clarity:** 3
**Significance:** 3
**Originality:** 3
**Rating:** 5
**Confidence:** 5

**Summary:**

The paper suggests a two-system model for continual learning of compositional tasks using RNNs.
In that, the what-system identifies the current context based on stimulus-response statistics
and the how-system learns to perform the individual task.
The how-system is implemented as an RNN and is modified by the what-system through re-weighting of its low-rank components.
Both systems are trained in parallel and online, such that the tasks are inferred while already learning in task.
The authors test their model on 6 related neuroscience tasks which involve matching or mirroring of a stimulus angle, memorising it and making decisions.
The tasks are formalised in a way that they consist of several epochs like fixation, stimulus, delay and response that can be inferred and composed.
Empirically their ContextRNN method outperforms other multi-task learning techniques (Elastic Weight Consolidation and Orthogonal Weight-Space Projection by Duncker 2020) in that it shows quicker context adaptation and less catastrophic forgetting.
Furthermore, they demonstrate that their model is able to do forward and backward transfer to related tasks as well as a simple example of compositional generalisation.

**Questions:**

1. How many low-rank components did you take, just 3? How would this affect the capacity of the RNN, in terms of how many tasks it can solve?
2. How many latents did the task-recognition network learn? Did that correspond to the number of epochs as defined in the task description?
3. How would your method compare to hyper-networks as used by Von Oswald et al.? Wouldn't this be a more fair comparison than EWC / OWP, as hypernetworks also separate the what and how?
3. How could your method be applied to settings with less structured tasks? Here you explicitly use tasks that are composed by clear epochs which are temporally separated. In the discussion you hint at non-parametric methods to find epochs. Could this also be trained in parallel to the how-RNN? (One would probably have to train in an interleaved instead of blocked-task regime then to learn what is similar between tasks.)

**Ethical Concerns:**

["NO or VERY MINOR ethics concerns only"]

**Final Justification:**

The authors have addressed most of my concerns, clarifying distinctions between tasks, the meaning of R, and R_M, and the conceptual (rather than mechanistic) nature of the biological plausibility claim. They provided helpful explanations of the low-rank decomposition, added relevant baselines, and improved clarity on “inference of tasks.” Though the small and closely related task set limits the scope of the findings, the compositional RNN approach to modeling cognitive tasks remains novel and serves as a proof of concept. Overall, the work is promising and invites future extensions to more diverse task compositions. I therefore decided to keep my score at 5.

**Limitations:**

Yes

**Quality:**

3

**Strengths And Weaknesses:**

- A strength of the paper is definitely the separation of task-recognition (what) and task-performance (how), which is not as clear-cut in many models. Most approaches use a single network to learn, such as EWC and OWP and therefore are less flexible.
-  Another strength is the method's ability to compose new tasks from similar known tasks. This is a benefit that also comes from the separate what-system, that recognises components of the tasks.
-  In my opinion, the choice of tasks is limited in that they are quite simple input-output mappings.
Especially the Delay and the Memory condition only differ in the duration of the delay but not in the input-output mapping (within Pro/Anti). This makes transfer from the Memory to the Delay domain very simple (as the RNN can use the same dynamics, just shorter). However, as one can see from Fig. S.2, the generalisation works less well for the other direction (again showing that the memory condition is harder).
- Clarity could be improved regarding the description of the what system. It took me a while to understand that it does not infer the context / task $c$ itself, but instead is given the context label c explicitly and only tries to infer the task component / epoch z. This is confusing in the introduction (e.g. line 40 "separation into contexts (what)") and only clarified later. Instead it might be called the "task decomposition" network.
-  Another point that needs explanation is the difference between the Delay conditions and the M' conditions.
-  They only seem to differ in the final epoch, which is just R in the Delay condition but R_M in the M' condition, although the M' is explicitly constructed without the memory epoch.
-  A minor weakness is the framing of the approach as brain-like. I agree that the brain is probably using a similiar separation of what and how. However, here the RNN needs to be decomposed into low-rank components, which is not a very neurally plausible operation (or could it be implemented in a more plausible way?) -- this caveat should be added to the discussion part that relates the method to multi-task learning in the brain.

---

> ### Author Rebuttal · Authors · 2025-07-30
>
> We thank the reviewer for their interest in our work and thoughtful comments. Below, we attempt to address the raised concerns, answer questions and highlight changes we have made in the revision in response to the reviews.
> 1. **Limited task set.** We agree that extending to a richer task set with more complex input-output mappings, with a more thorough assessment of the potential for (and directionality of) transfer learning and compositionality, is important. At the present stage, we focus on synthetic problems common in neuroscience models for interpretability and relevance for understanding Continual Learning and compositional re-use in RNNs. In related work studying similarly simple tasks (e.g. refs. 11, 13-16), compositional reuse in multi-tasking RNNs has been of major interest. Our work is intended to contribute to this space, making the compositional task structure, and how it may shape a potential RNN solution, an explicit focus. We note that our approach for modeling the statistical structure of cognitive tasks in a cohesive generative framework is novel. We expect that this contribution will help to establish a more formal notion of task-similarity and when/how transfer might be simple or more difficult. We fully agree that a richer task set will lead to a variety of interesting questions for further work.
> 2. **Writing in Sec. 3.** We thank the reviewers for suggestions about improving the readability of Sec. 3, which was a concern also shared by other reviewers. We revised the writing of Sec. 3 to make it more accessible to readers unfamiliar with probabilistic modeling. In particular, we made sure to highlight conceptual features of the algorithm first, before introducing formal notation. We also stressed that the model is given the correct task label $c$ and only performs the epoch inference $z$. If there are additional parts of Sec. 3 that the reviewer found to be particularly inaccessible to the reader, we would appreciate more detail (e.g. line numbers), so we can make sure to improve their clarity.
> 3. **Differentiating Memory and Delay tasks.** We have updated the wording to clarify that there are two differences between these sets of tasks. First, the Memory tasks have a fixation epoch after stimulus presentation. Second, the correct stimulus is present during the response epoch for Delay tasks but not during the response epoch for Memory tasks (see Fig. 1a). We have added details on this to section 4.4.
> 4. **Biological plausibility.** We agree that the current version of the architecture lacks biological realism and is only brain-like at a conceptual level. We are very interested in exploring more biologically plausible architectures that could implement the modulation of low-rank components of the RNN. [16] connected a similar selection of low-rank components to neuromodulation. [37] implemented the selection of individually controlled low-rank components through an architecture representing interactions between basal ganglia, thalamus and cortex. This model assumes that the low-rank connectivity of the RNN is the result of cortex-to-thalamus and thalamus-to-cortex connections (= thalamo-cortical ‘loop’). When the number of thalamic neurons participating in each loop is small, the connectivity is low rank. Different loops can be selected via inhibitory input from basal ganglia to thalamus, which effectively turns on and off different low-rank components of the RNN – just like our architecture. Contrary to our work, [37] was not concerned with how this inhibitory selection signal is set.
> 5. **Low-rank components (Q1).** The number of low-rank weight matrices (a set of low-rank components) equals the number of unique task epochs. For our current task set, the inference algorithm identifies 9 unique task epochs in total. Each task epoch uses a rank-3 recurrent weight matrix in our main figures (defined by $U_z, V_z$, Appendix C.1), though performance increased for higher ranks (Figure S5). For task-epochs solved by an at most rank-$R$ weight matrix, we may implement $N_z = N/R$ epochs for a fixed network dimensionality $N$. The network can solve tasks that can be composed as sequences of the $N_z$ epochs, which could be a large number. It is worth noting that, since each component of the network uses low-rank recurrent weights, the memory cost of adding a new component is linear in RNN size, as opposed to the quadratic scaling in vanilla RNNs. To improve scalability, a possible next step is to discourage the addition of new epochs by penalizing having a large $N_z$. Alternatively, it may also be possible to group epochs that are statistically distinct, but can share the same solution through another learning mechanism. This would be particularly relevant for the response epochs in our task set.
> 6. **Number of learned latents (Q2).** Across the 6 tasks we studied, 10 epochs are defined as part of the generative process of the tasks (listed in line 93 of the appendix). The ‘what’ system learns this generative model based on the inputs and targets of each task, and identifies only 9 epochs. This is expected, as the fixation epoch $F$ and memory epoch $M$ are indistinguishable based on their input-output patterns.
> 7. **Additional baselines (Q3).** We agree with the reviewer that a comparison against the hypernetwork approach in von Oswald et al. [23] and Ehret et al. [31] is warranted. We have performed these comparisons and are planning to include them in the revised manuscript. Per the suggestion of other reviewers, we also performed a comparison against the neuromodulatory network in Costacorta et al., 2024 [16]. In short, we found that neither method matched the performance of ours in terms of continual learning or compositional generalization. Details and performance metrics are provided at the end of this reply.
> 8. **Less structured tasks (Q4).** Handling tasks with less apparent structures is an important next step. Training the 'what' system together with the 'how' RNN is indeed a promising approach. In the case of ambiguous structures, it may be helpful to use the hidden state of the 'how' RNN as part of the observation $\mathbf{q}_t$ to infer epochs. This is similar in spirit to the scheme in Ostapenko et al., 2021 for modules in a feedforward network, where the output of each module is used to determine its relevance for a given input.
> ***
> **Summary of additional baselines.** Costacurta et al., 2024 [16] designed neuromodulated RNN (NM-RNN), in which a full-rank modulatory RNN (RNN1) outputs a low-D signal that dynamically scales the low-rank recurrent weights of an output-generating RNN (RNN2). RNN1 and RNN2 are jointly optimized through gradient descent. Their design showed catastrophic forgetting during continual learning (Table 1). In contrast, our method replaces RNN1 with a probabilistic task model, which rapidly learns a task’s structure within a handful of trials, thus is able to provide useful contextual signals when optimizing RNN2 and facilitate continual learning. Von Oswald et al., 2020 [23] designed task-conditioned hypernetworks to tackle continual learning. Their hypernetwork receives a learnable embedding distinct for each task and outputs the weights for the target network (a RNN in our case). Their design underperformed our method (Table 1). More importantly, these hypernetworks do not capture the time-varying epoch structures of each task, and perform worse on compositional generalization (Table 2).
>
> **Implementation details.**
>
> **NM-RNN**:  RNN2 has 256 neurons and with rank($W^{rec}$) = 27, equal to the default choice for our ContextRNN. RNN1 has 125 neurons such that the whole network has a similar number of parameters as ours.
>
> **Hypernetwork**:  The full-rank RNN has 256 neurons. We used a chunked hypernetwork with 34 chunks and 2000-D output. The hypernetwork is a MLP with two 32-D hidden layers, receiving 32-D task embedding and chunk embeddings as input. After a grid search, we set the continual learning regularization strength to 1, orthogonal regularization strength to 0.001 and clip the gradient norm to 1 (see Ehret et al., 2021).
>
> Table 1:
>
> **A. NM-RNN**
>
> | | task 1| task 2| task 3| task 4| task 5| task 6|
> |-|-|-|-|-|-|-|
> | final perf.| 0.00 (± 0.00)| 0.25 (± 0.06)| 0.00 (± 0.00)| 0.15 (± 0.05)| 0.00 (± 0.00)| 1.00 (± 0.00)|
>
> **B. Hypernetwork**
>
> | | task 1| task 2| task 3| task 4| task 5| task 6|
> |-|-|-|-|-|-|-|
> | final perf.| 0.99 (± 0.01)| 0.89 (± 0.18)| 0.79 (± 0.06)| 0.67 (± 0.11)| 0.80 (± 0.15)| 0.87 (± 0.06)|
>
> **C. ContextRNN (ours)**
>
> | | task 1| task 2| task 3| task 4| task 5| task 6|
> |-|-|-|-|-|-|-|
> | final perf.| 0.99 (± 0.02)| 0.99 (± 0.02)| 0.91 (± 0.04)| 0.93 (± 0.04)| 1.00 (± 0.00)| 1.00 (± 0.00)|
>
> Caption: Continual learning performance (supplementary to Figure 4 a-d). A: Each column shows a task's final performance averaged across 5 random seeds (± std) after training NM-RNN sequentially on task 1 to 6. B: same as A but for hypernetwork. C: same as A but for contextRNN (ours).
>
> Table 2:
>
> **A. OWP**
>
> | | after 0 trials | after 20 trials | after 40 trials | after 512 trials |
> | - | - | - | - | - |
> | MemoryAnti perf. | 0.00 (± 0.00)  | 0.00 (± 0.01)   | 0.01 (± 0.02)   | 0.53 (± 0.09)    |
> **B. Hypernetwork**
>
> | | after 0 trials | after 20 trials | after 40 trials | after 512 trials |
> | - | - | - | - | - |
> | MemoryAnti perf. | 0.00 (± 0.00)  | 0.03 (± 0.02)   | 0.19 (± 0.05)   | 0.64 (± 0.03)    |
> **C. ContextRNN (ours)**
>
> | | after 0 trials | after 20 trials | after 40 trials | after 512 trials |
> | - | - | - | - | - |
> | MemoryAnti perf. | 0.00 (± 0.00)  | 0.56 (± 0.06)   | 0.83 (± 0.02)   | 0.83 (± 0.04)    |
>
> Caption: Compositional generalization (supplementary to Figure 6c). A: Each column shows the test performance on MemoryAnti averaged across five random seeds (± std) after training a pretrained RNN using OWP on MemoryAnti for 0, 20, 40 or 512 trials. B: same as A but for hypernetwork. C: same as A but for contextRNN (ours).

---

> ### Comment · Reviewer_yJUA · 2025-08-05
> **thanks for the thorough rebuttals!**
>
> I thank the authors for responding thoroughly to all the raised points and for providing additional baselines. Specifically, I appreciate the following:
>
> 1. The authors address that the experiment setup is a bit limited in that it has a small set of tightly related tasks. I agree that modelling the cognitive tasks using RNNs in this compositional way is novel and serves as proof of concept.
> 2. They clarified the writing wrt to the "inference of tasks", as opposed to episodes.
> 3. I might have misunderstood the difference in R and R_M, which is in whether the stimulus is still on screen. Thanks for the clarification.
> 4. They addressed biological plausibility and clarified that it only exists on a conceptual level. Interesting that low-rank decomposition could be implemented, I should check out (37). It may be cool for neuroscientists to test in neural data, if there is what and how networks in the brain interact with these low-rank components.
> 5. Thanks for clarifying how the decomposition works and the number of latents.
> 6. I appreciate the additional baselines, especially the hypernetwork comparison. Cool that your method comes out better across tasks.
>
> However, some points are still puzzling me.
> I understand the differences between Memory and Delay task now (which is the additional fixation period and the stimulus visibility), but I would still argue that very similar dynamics can be used in how networks, so the improved transfer between the two seems not very surprising. The same holds for the M' task, which is quite close to the Delay task.
> In the future, it would be interesting to see how generalization to other possible compositions of the epochs behaves. E.g., one could imagine a hard task having 2 stimuli of which only the first has to be considered for the answer.
>
> I keep my overall score at 5.

---

> > ### Author Response · Authors · 2025-08-06
> > **Thank you**
> >
> > We thank the reviewer for the appreciation of our work. Regarding the comment on transferability between Memory and Delay tasks -- indeed, given the similarity between these tasks, a model with good compositionality should share dynamics in solving them. Indeed, previous work in Driscoll et al. (2024) and Costacurta et al (2024) has shown that RNNs can exploit this similarity in solving the tasks. However, this has not been previously demonstrated while sequentially learning the tasks, and by making the composition (and similarity) of different computational building-blocks explicit via contextual inference. Regarding the experiments with two stimuli coming on at different times, we have actually implemented this in our codebase already. and are testing it in ongoing work.

---

### Official Review · Reviewer_N8zU · 2025-06-26

**Clarity:** 1
**Significance:** 2
**Originality:** 2
**Rating:** 4
**Confidence:** 4

**Summary:**

This paper investigates how neural systems can continually learn and reuse compositional skills to solve a variety of tasks, especially when those tasks share common structural elements but differ in specific details. The major challenge is to avoid catastrophic forgetting and to enable generalization by recombining learned components. To address this, the authors introduce a two-system framework that explicitly separates "what" (context inference) from "how" (computation implementation). The what system employs a probabilistic generative model to segment tasks into discrete epochs and learns to infer these contexts online, while the how system use an RNN whose weights are dynamically modulated via low-rank components according to the inferred context from the what system. Experiments on a set of neuroscience-inspired tasks demonstrate that this approach effectively mitigates catastrophic forgetting and supports compositional generalization.

**Questions:**

Please refer to earlier sections.
Minor question: Is it possible to replace the RNN in the “how” component with other types of models? For example, could the proposed framework work equally well if the RNN were substituted with a feedforward neural network, a transformer, or another architecture more suitable for the task at hand? It would be helpful if the authors could clarify whether their approach is generally applicable to other model classes beyond RNNs, and discuss any potential challenges or limitations in doing so.

**Ethical Concerns:**

["NO or VERY MINOR ethics concerns only"]

**Final Justification:**

Most of my concerns have been addressed through the addition of extended baseline comparisons and improved discussion of assumptions and limitations. Therefore, I increased my score.

**Limitations:**

As mentioned in the weaknesses above, the paper would be strengthened by (1) evaluating its applicability to domains without clear compositional task structure, (2) providing ablation studies on the sensitivity to context inference design and ambiguous context boundaries, (3) more thoroughly distinguishing its approach from related work in meta-learning, hypernetworks, and neuromodulation, (4) including a broader set of baselines, particularly recent modular and task-inference-based continual learning methods, and (5) improving the clarity of Section 3 for readers.

**Paper Formatting Concerns:**

None.

**Quality:**

2

**Strengths And Weaknesses:**

Strengths
1. Comprehensive experiments: The experiments are thorough and cover catastrophic forgetting, transfer, and compositional generalization.
2. Originality: While related ideas exist, the explicit and interpretable separation of "what" and "how" in continual learning RNNs is new and well-justified.

Weaknesses
1. Assumptions on task structure: The probabilistic model assumes that tasks can be decomposed into discrete, shared epochs. This may limit applicability to domains where such structure is not present or easily defined.
2. Ablation on module interdependence: There is little analysis of how sensitive the approach is to the precise design of the context inference model, or what happens if context boundaries are ambiguous or misspecified.
3. Related work overlap: While the two-system approach is novel, there is some overlap with prior work in meta-learning, hypernetworks, and neuromodulation; the paper could do more to clarify distinctions and advantages over these related methods.
4. Limited baselines: The experiments do not include comparisons to more recent or diverse modular, meta-learning, or task-inference-based continual learning frameworks (e.g. Costacurta et al. Structured flexibility in recurrent neural networks via neuromodulation. Advances in Neural Information Processing Systems, 2024). This limitation makes it difficult to fully assess the relative advantages of the proposed approach.
5. Clarity: Section 3 is dense and may be challenging for readers unfamiliar with probabilistic modeling.

---

> ### Author Rebuttal · Authors · 2025-07-30
>
> We thank the reviewer for their interest in our work and thoughtful comments, and appreciate that the reviewer thought our submission makes a novel contribution that was thoroughly demonstrated through experiments and evaluations. The explicit suggestions for how to improve the paper were very helpful, and we are making the following changes in response to them:
> 1. **Additional baselines & more thorough discussion of related work.** We agree with the reviewer that additional baseline comparisons are necessary and have added two additional baselines in the manuscript. The first one is against the recent Costacurta et al., 2024 [16] neuromodulatory network and a second is against a hypernetwork approach (von Oswald et al., 2020, [23]). In short, we found that neither method matched the performance of ours in terms of continual learning or compositional generalization. Details and performance metrics are provided at the end of this reply. We have updated the 'Related work and our contributions' section to discuss these works further and also expanded the baseline comparisons (Fig. 4) to include these two. We hope that these changes will address Limitations (3) and (4).
> 2. **Improved clarity of Sec. 3.** We revised the writing of Sec. 3 to make it more accessible to readers unfamiliar with probabilistic modeling. In particular, we made sure to highlight conceptual features of the algorithm first, before introducing formal notation. We also stressed that the model is given the correct task label $c$ and only performs the epoch inference $z$. If there are parts of Sec. 3 that the reviewer found to be particularly inaccessible to the reader, we would appreciate more detail e.g. line numbers, so we can make sure to improve their clarity. We hope that these changes will address Limitation (5).
>
> We respond to the other concerns below.
>
> 3. **Assumptions on task structure.** This concern is related to the assumptions about task structures that we make in the paper, and raises the question of whether our approach is applicable to domains without clear compositional task structures. At the present stage, we have chosen to focus on synthetic problems common in neuroscience modeling work for interpretability and relevance to understanding continual learning in the brain. As stated in line 107-108, we decided to focus on common experimental paradigms, which are often modeled as having clear boundaries between epochs (see refs. on line 108). In related literature (e.g. [11], [13]-[16]), compositional reuse in multi-tasking RNNs has been of major interest. Our work is intended to contribute to this space, making the compositional task structure, and how it may shape a potential solution in the RNN, an explicit focus. Still, we wholeheartedly agree with the reviewer that it is important to evaluate our approach on more realistic tasks, spanning a wider domain. Analyzing the approach in settings with more ambiguous compositional structures will be particularly interesting. While we argue that this is beyond the scope of the current paper, we would like to highlight that the probabilistic treatment of the 'what' system may be especially well-equipped to handle these settings.
> 4. **Ablations and ambiguous context boundaries.** The 'what' system is designed to perform posterior inference of context, conditioned on the observables (inputs $\mathbf{s}_t$ and targets $\mathbf{y}_t$) of a given task, and, in principle, any architecture that can do this could be used as the context inference model. In our setting,  ambiguity across epoch boundaries stems only from the temporal variability of each trial and observation noise. We agree that it would be worthwhile to further explore the setting where context boundaries are ambiguous due to other sources. For instance, this could happen in our tasks when stimuli change more slowly (e.g. when they are supplied as a ramping signal rather than turning on/off abruptly). Studying this task variation would test how robust our inference system is to a mismatch between the underlying generative assumptions, and the true task design, which is important. Given the uncertainty about context that would arise at these transition points, the 'how' system would express a mixture of weights rather than a single solution. We would expect that this would impact the task performance, and might encourage (or require) learning some form of normalization across weights. Currently, the system only experiences context-uncertainty during learning. We are planning to update the discussion section to include these points. Regarding the suggested ablation studies, we would appreciate additional detail on what exactly the reviewer had in mind.
> 5. **Replacing the RNN with other models.** The general framework of using probabilistic inference to 'decompose' time-series tasks is agnostic to the 'how' module used. Indeed we imagine a similar scheme to work for other architectures that deal with time series, such as transformers. For example, with transformers the equations in Eq. 3-5 would be replaced by analogous equations that determine the query, key, value weights used for input from different epochs. Incorporating the two-systems approach we develop here into more complex architectures represents an interesting direction of future work and may also facilitate further applications to more complex tasks.
> ***
> **Summary of additional baselines.** Costacurta et al., 2024 [16] designed neuromodulated RNN (NM-RNN), in which a full-rank neuromodulatory RNN (RNN1) outputs a low-D signal that dynamically scales the low-rank recurrent weights of an output-generating RNN (RNN2). RNN1 and RNN2 are jointly optimized through gradient descent. Their design showed catastrophic forgetting during continual learning (Table 1). In contrast, our method replaces RNN1 with a probabilistic task model, which rapidly learns a task’s structure within a handful of trials, thus is able to provide useful contextual signals when optimizing RNN2 and facilitate continual learning. Von Oswald et al., 2020 [23] designed task-conditioned hypernetworks to tackle continual learning. Their hypernetwork receives a learnable embedding distinct for each task and outputs the weights for the target network (a RNN in our case). Their design underperformed our method (Table 1). More importantly, these hypernetworks do not capture the time-varying epoch structures of each task, and performed worse on compositional generalization (Table 2).
>
> **Implementation details.**
>
> **NM-RNN**:  RNN2 has 256 neurons and with rank($W^{rec}$) = 27, equal to the default choice for our ContextRNN. RNN1 has 125 neurons such that the whole network has a similar number of parameters as ours.
>
> **Hypernetwork**:  The full-rank RNN has 256 neurons. We used a chunked hypernetwork with 34 chunks and 2000-D output. The hypernetwork is a MLP with two 32-D hidden layers, receiving 32-D task embedding and chunk embeddings as input. After a grid search, we set the continual learning regularization strength to 1, orthogonal regularization strength to 0.001 and clip the gradient norm to 1 (see Ehret et al., 2021 for details).
>
> Table 1:
>
> **A. NM-RNN**
>
> || task 1| task 2| task 3| task 4| task 5| task 6|
> |-|-|-|-|-|-|-|
> | final perf.| 0.00 (± 0.00)| 0.25 (± 0.06)| 0.00 (± 0.00)| 0.15 (± 0.05)| 0.00 (± 0.00)| 1.00 (± 0.00)|
>
> **B. Hypernetwork**
>
> || task 1| task 2| task 3| task 4| task 5| task 6|
> |-|-|-|-|-|-|-|
> | final perf.| 0.99 (± 0.01)| 0.89 (± 0.18)| 0.79 (± 0.06)| 0.67 (± 0.11)| 0.80 (± 0.15)| 0.87 (± 0.06)|
>
> **C. ContextRNN (ours)**
>
> || task 1| task 2| task 3| task 4| task 5| task 6|
> |-|-|-|-|-|-|-|
> | final perf.| 0.99 (± 0.02)| 0.99 (± 0.02)| 0.91 (± 0.04)| 0.93 (± 0.04)| 1.00 (± 0.00)| 1.00 (± 0.00)|
>
> Caption: Continual learning performance (supplementary to Figure 4 a-d). A: Each column shows a task's final performance averaged across 5 random seeds (± std) after training NM-RNN sequentially on task 1 to task 6. B: same as A but for hypernetwork. C: same as A but for contextRNN (ours).
>
> Table 2:
>
> **A. OWP**
>
> |                  | after 0 trials | after 20 trials | after 40 trials | after 512 trials |
> | ---------------- | -------------- | --------------- | --------------- | ---------------- |
> | MemoryAnti perf. | 0.00 (± 0.00)  | 0.00 (± 0.01)   | 0.01 (± 0.02)   | 0.53 (± 0.09)    |
> **B. Hypernetwork**
>
> |                  | after 0 trials | after 20 trials | after 40 trials | after 512 trials |
> | ---------------- | -------------- | --------------- | --------------- | ---------------- |
> | MemoryAnti perf. | 0.00 (± 0.00)  | 0.03 (± 0.02)   | 0.19 (± 0.05)   | 0.64 (± 0.03)    |
> **C. ContextRNN (ours)**
>
> |                  | after 0 trials | after 20 trials | after 40 trials | after 512 trials |
> | - | - | - | - | - |
> | MemoryAnti perf. | 0.00 (± 0.00)  | 0.56 (± 0.06)   | 0.83 (± 0.02)   | 0.83 (± 0.04)    |
>
> Caption: Compositional generalization (supplementary to Figure 6c). A: Each column shows the test performance on MemoryAnti averaged across five random seeds (± std) after training a pretrained RNN using OWP on MemoryAnti for 0, 20, 40 or 512 trials. B: same as A but for hypernetwork. C: same as A but for contextRNN (ours).

---

> > ### Comment · Reviewer_N8zU · 2025-08-04
> >
> > I appreciate the authors' detailed and thoughtful responses. Most of my concerns have been addressed through the addition of extended baseline comparisons and improved discussion of assumptions and limitations. These clarifications and results should be clearly reflected in the revised manuscript. Regarding response 5, I believe that including a basic experiment to test robustness under ambiguous transitions (e.g., via added noise or smoothed stimuli) would significantly strengthen the claim of generalizability and broaden the potential applicability of the proposed framework.
> >
> > I am happy to increase my score to 4.

---

> > > ### Author Response · Authors · 2025-08-06
> > > **Thank you**
> > >
> > > We thank the reviewer for the appreciation of our work and our responses. We will be careful in making sure that the revised manuscript reflects all clarifications and additional results. The suggested experiment about robustness under ambiguous transitions is interesting. We will include it as part of our ongoing work of improving the generalizability of this model.

---

### Official Review · Reviewer_jVxy · 2025-06-26

**Clarity:** 3
**Significance:** 2
**Originality:** 3
**Rating:** 4
**Confidence:** 4

**Summary:**

This manuscript proposed to use the two-stream (what and how) processing idea from the visual system as a way to efficiently solve the continual learning problem. The 'what' component is an RNN trained with a large family of tasks to obtain the compositional latents (vocabulary) of tasks for generalization. The 'how' component is a low-rank RNN conditioned on the context inferred from the 'what' system. The epoch variable z and trial variable x as the latent contexts for six cognitive task. The weights to each epoch are distinct, making the network more flexible for different epochs. The authors also proposed a task model and use EM algorithm to infer the latent contexts, guiding the parameter selection of the downstream low-rank RNN. In the comparison experiment with two traditional methods in continual learning (EWC and OWP), the proposed method shows better continual learning ability and exhibits positive forward/backward transfer effect. The reusability of the epoch context has also been verified. In summary, this article provides new insights for the continual learning of RNNs.

**Questions:**

Opportunities for improvement
1. More comprehensive comparison. The strength of the 'epoch' concept could be further demonstrated by adding comparison experiments with reference [16], as well as with the case where the weights are modulated by task context c instead of epoch context z.
2. More detailed explanation. Figure 5b shows that at the beginning of the training process of the second task, the test loss of the second task will first increase and then decrease. The authors are suggested to provide an explanation about that.
3. Clarification: The image descriptions can be improved to clarify that the curves of different colors indicate the test performance on the corresponding tasks and the background color indicate the training process of the corresponding tasks. In appendix the stimulus angle in section A and the model parameters in section B are both denoted as theta, which may cause potentially confusion.

**Ethical Concerns:**

["NO or VERY MINOR ethics concerns only"]

**Final Justification:**

I have updated the final rating to 4.

**Limitations:**

Yes

**Quality:**

2

**Strengths And Weaknesses:**

Strength:
1. Explicit compositional reuse. This paper separates the 'what' model to infer latent contexts rather than training the whole model in an end-to-end way, making  the compositional reuse of RNN components across contexts more explicit.
2. Efficient network structure. The recurrent weights of RNN are represented as low-rank matrices, reducing the number of parameters. The proposed context-modulated RNN uses half as many trainable parameters to achieve comparative performance to traditional methods.
3. Probabilistic perspective. This paper utilizes weights of different contexts from a probabilistic perspective, combining them in a linear manner weighted by the belief instead of picking up weights from one context, making the model more flexible.

Weaknesses:
1. Contradiction between the precision and reusability of the epoch. In this paper, on the one hand, the response period of the six tasks are all defined with distinct epochs in order to model the neural activities more accurately, sacrificing reusability of epoch. While on the other hand, defining stimulus periods of all tasks as the same epoch makes it difficult for the model to capture the different dynamics of the tasks during stimulus period, such as the evidence accumulation phenomenon.
2. Discontinuous dynamics in a trial. By assigning different weights for different epoch, the dynamics of the RNN in a trial is separated by epochs, making it impossible to simulate dynamics such as sub-threshold evidence accumulation process. As a consequence the ability of the model is largely limited especially faced with complex tasks.
3. Limited supported task types. According to the appendix, x determines the stimulus angle, a feature that carries consistent meaning across tasks. As a result, the method is constrained to tasks involving stimuli with angular information, thereby limiting its applicability to a broader range of task types. Additionally, Figure 2c exhibits multiple stimulus-response pairs under a single x within a task, which raises questions about the uniqueness and interpretability of the mapping.
4. Practicality. The data for the six cognitive tasks is generated under relatively simple and controlled rules, which may differ from the distribution of real biological data. This raises concerns regarding the ability of the proposed method to capture the underlying neural dynamics present in real organisms.

Significance: The question of continual learning is an unresolved topic, leveraging the concept of compositionally and efficiency is of importance. However, the work itself still need more training dataset, testing and model comparison for evaluation.
Originality：The concept to separate into "what" and "how" models, and the weight composition in a probabilistic perspective are relatively novel.
Clarity：This article is well-written, with a complete introduction to relevant concepts and works and a detailed description about the proposed what and how models.
Quality: The results are not comprehensive enough, lacking controls and oblation study. The proposed method was only compared with the traditional EWC and OWP methods, without comparing with more similar methods like the reference [16], making the comparison experiment less persuasive. Furthermore, the six cognitive tasks are most distinguishable during their response periods. However, the response periods of these tasks are all treated as distinct epochs, making the reusability of the epoch context seem less meaningful. In addition, the data used for the six tasks are simulated, raising concerns about the real-world applicability of the proposed method.

---

> ### Author Rebuttal · Authors · 2025-07-30
>
> We appreciate that the reviewer thought that we made novel contributions to addressing an important open problem in the space of continual learning, and that the probabilistic treatment of tasks and context inference is a strength of the paper. The explicit suggestions for how to improve the paper and address the concerns are very helpful. In response, we have made the following changes:
> 1. **Additional baselines.** We have now included a comparison against the NM-RNN [16], and (suggested by another reviewer) hypernetworks [23, 31]. Neither method matched the performance of ours in terms of continual learning or compositional generalization. Details and performance metrics are provided at the end of this reply. We have updated Sec. 2 to discuss these works further and also expanded the baseline comparisons (Fig. 4) to include these two.
> 2. **Explanation for Fig. 5b.** The degree of loss increase varied across random seeds, and happened after the first step of gradient descent on the second task (our batch size equals 256). It could stem from the discrepancy between the first training batch and the test dataset, or that the model moved away from a local minimum of the second task's loss landscape.
> 3. **Figure captions.** cf. 'The image descriptions … the corresponding tasks.' We have incorporated this clarification into our captions.
>
> Other concerns:
>
> 4. **Contradiction between the precision and reusability of the epoch.** This point seems to refer to the problem of identifying the correct segmentation of tasks into epochs. As the reviewer noted, it is possible to share stimulus periods, and/or response periods depending on how exactly tasks are segmented. Indeed, for a subset of the tasks we consider here (Memory/Delay Pro/Anti) both segmentations are equivalent. The reviewer’s concern, to our understanding, is that we chose a particular segmentation to better model neural activity, at the sacrifice of re-usability of epochs across tasks. To clarify: we infer the segmentation based only on the statistical structure of the task, not neural activity. In addition to this, sharing of the stimulus epoch only happens for tasks that share the same statistical structure during the stimulus epoch, and therefore does not limit the system’s ability to capture different dynamics when this is necessitated by the demands of the task. For example, the stimulus periods between the DM tasks, and the Delay/Memory tasks are not inferred to be shared. More generally, the tension between oversegmentation and undersegmentation is an issue common to mixture models (e.g., clustering, hidden Markov models), and our approach is no exception. Further clarification on this point by the reviewer would be much appreciated.
> 5. **Discontinuous dynamics in a trial.** This point is concerned with dynamics of the RNN and whether the model can capture 'dynamics such as sub-threshold evidence accumulation process'. While the inputs and target outputs are piecewise constant and sometimes discontinuous between epochs, the RNN hidden state (Eq. 1) is not constrained in these ways. The network can only respond correctly during response epochs using information contained in its hidden state. Within each epoch (for example during a stimulus epoch) it may exhibit ramping dynamics that are signatures of evidence accumulation. Our DM tasks are designed to emulate evidence accumulation: the network has to integrate noisy inputs to make a decision relating to the stimulus strength. Our choice of piecewise constant input plus noise follows classic models of evidence accumulation in the literature (e.g., Wang, 2002; Ratcliff et al., 2016). In summary, while the weights of the network switch across epochs, the network state evolves continuously and can exhibit complex activity patterns.
> 6. **Limited supported task types.** The reviewer was concerned that our choice of $x$ constrains us to tasks involving stimuli with angular information. Our approach, however, does not require a consistent interpretation of $x$ across tasks. In fact, among the 6 tasks we studied, $x$ determines the stimulus angle in only 4 of them (Delay/Memory Pro/Anti). In the two DM tasks, $x$ controls the stimulus location/strength (appendix lines 94-100). Thus, our task set demonstrates the applicability to a broader range of tasks. Regarding the comment on Fig. 2c ('multiple stimulus-response pairs under a single x within a task'): while $x$ is a single discrete variable taking on a single value per trial of a given task, the stimulus ($\mathbf{s}_t$) and response variable ($\mathbf{y}_t$) can be higher-dimensional, as is the case in our example (both are 3D, corresponding to the three lines in the figure). We apologize for the confusion and have updated the figure to clarify this.
> 7. **Practicality.** This point is concerned with the generation of the cognitive tasks and whether they can match the distribution of real biological data. While we are excited about connecting our model to neural data in the future, we want to clarify that at the moment this work is not concerned with modeling biological neural activity at all. The synthetic data, represented by $\mathbf{q}_t$, are meant to model the behavior-level input (stimuli on a screen) and output (motor actions, eye saccades) in neuroscience experiments. The simplified dynamics, where the inputs are piecewise constant, are consistent with typical experimental setups and are common assumptions in related modeling work (see refs. on line 108).
> 8. The reviewer suggested comparing to the setting where weights are modulated by task context c instead of epoch context z. This would effectively train a separate network per task, and therefore sidestep both the continual learning problem and any compositional reuse across tasks. This would make forward/backward transfer impossible. We are happy to include these comparisons, but would appreciate further clarification on what the reviewer would like us to show.
>
> **References:**
>
> Wang, X. J. (2002). Neuron, 36(5), 955-968.
> Ratcliff, R., ..., & McKoon, G. (2016). Trends in cognitive sciences, 20(4), 260-281.
> ***
> **Summary of additional baselines.** Costacurta et al., 2024 [16] designed neuromodulated RNN (NM-RNN), in which a full-rank neuromodulatory RNN (RNN1) outputs a low-D signal that dynamically scales the low-rank recurrent weights of an output-generating RNN (RNN2). RNN1 and RNN2 are jointly optimized through gradient descent. Their design showed catastrophic forgetting during continual learning (Table 1). In contrast, our method replaces RNN1 with a probabilistic task model, which rapidly learns a task’s structure within a handful of trials, thus is able to provide useful contextual signals when optimizing RNN2 and facilitate continual learning. Von Oswald et al., 2020 [23] designed task-conditioned hypernetworks to tackle continual learning. Their hypernetwork receives a learnable embedding distinct for each task and outputs the weights for the target network (a RNN in our case). Their design underperformed our method (Table 1). More importantly, these hypernetworks do not capture the time-varying epoch structures of each task, and perform worse on compositional generalization (Table 2).
>
> **Implementation details.**
>
> **NM-RNN**:  RNN2 has 256 neurons and with rank($W^{rec}$) = 27, equal to the default choice for our ContextRNN. RNN1 has 125 neurons such that the whole network has a similar number of parameters as ours.
>
> **Hypernetwork**:  The full-rank RNN has 256 neurons. We used a chunked hypernetwork with 34 chunks and 2000-D output. The hypernetwork is a MLP with two 32-D hidden layers, receiving 32-D task embedding and chunk embeddings as input. After a grid search, we set the continual learning regularization strength to 1, orthogonal regularization strength to 0.001 and clip the gradient norm to 1 (see Ehret et al., 2021 for details).
>
> Table 1:
>
> **A. NM-RNN**
>
> || task 1| task 2| task 3| task 4| task 5| task 6|
> |-|-|-|-|-|-|-|
> | final perf.| 0.00 (± 0.00)| 0.25 (± 0.06)| 0.00 (± 0.00)| 0.15 (± 0.05)| 0.00 (± 0.00)| 1.00 (± 0.00)|
>
> **B. Hypernetwork**
>
> || task 1| task 2| task 3| task 4| task 5| task 6|
> |-|-|-|-|-|-|-|
> | final perf.| 0.99 (± 0.01)| 0.89 (± 0.18)| 0.79 (± 0.06)| 0.67 (± 0.11)| 0.80 (± 0.15)| 0.87 (± 0.06)|
>
> **C. ContextRNN (ours)**
>
> || task 1| task 2| task 3| task 4| task 5| task 6|
> |-|-|-|-|-|-|-|
> | final perf.| 0.99 (± 0.02)| 0.99 (± 0.02)| 0.91 (± 0.04)| 0.93 (± 0.04)| 1.00 (± 0.00)| 1.00 (± 0.00)|
>
> Caption: Continual learning performance (supplementary to Figure 4 a-d). A: Each column shows a task's final performance averaged across 5 random seeds (± std) after training NM-RNN sequentially on task 1 to task 6. B: same as A but for hypernetwork. C: same as A but for contextRNN (ours).
>
> Table 2:
>
> **A. OWP**
>
> |                  | after 0 trials | after 20 trials | after 40 trials | after 512 trials |
> | ---------------- | -------------- | --------------- | --------------- | ---------------- |
> | MemoryAnti perf. | 0.00 (± 0.00)  | 0.00 (± 0.01)   | 0.01 (± 0.02)   | 0.53 (± 0.09)    |
> **B. Hypernetwork**
>
> |                  | after 0 trials | after 20 trials | after 40 trials | after 512 trials |
> | ---------------- | -------------- | --------------- | --------------- | ---------------- |
> | MemoryAnti perf. | 0.00 (± 0.00)  | 0.03 (± 0.02)   | 0.19 (± 0.05)   | 0.64 (± 0.03)    |
> **C. ContextRNN (ours)**
>
> |                  | after 0 trials | after 20 trials | after 40 trials | after 512 trials |
> | - | - | - | - | ---------------- |
> | MemoryAnti perf. | 0.00 (± 0.00)  | 0.56 (± 0.06)   | 0.83 (± 0.02)   | 0.83 (± 0.04)    |
>
> Caption: Compositional generalization (supplementary to Figure 6c). A: Each column shows the test performance on MemoryAnti averaged across five random seeds (± std) after training a pretrained RNN using OWP on MemoryAnti for 0, 20, 40 or 512 trials. B: same as A but for hypernetwork. C: same as A but for contextRNN (ours).

---

> > ### Comment · Reviewer_jVxy · 2025-08-04
> >
> > Based on the rebuttal, I would be happy to increase the overall rating to 4.

---

> > > ### Author Response · Authors · 2025-08-06
> > > **Thank you**
> > >
> > > We thank the reviewer for the appreciation of our work and raising the score. If there are further questions and concerns, feel free to let us know.

---

### Official Review · Reviewer_fkcB · 2025-06-30

**Clarity:** 4
**Significance:** 3
**Originality:** 3
**Rating:** 5
**Confidence:** 4

**Summary:**

The authors focus on continual learning in RNNs.

Specifically on compositional computations and modular reuse across sequentially learned tasks, which in my opinion is a topic of increasing significance.

The paper introduces a dual-system architecture consisting of a "what" system, which probabilistically infers computational contexts or epochs from input data streams, and a "how" system, which dynamically modulates RNN parameters based on the inferred context.

The paper has a  clean theoretical formulation, good interpretability through structured context inference, and it gives solid empirical evidence to demonstrate superior continual learning and rapid generalization to novel compositional tasks.

**Questions:**

1. is it possible to include a more realistic task that comes from an NLP real-world application in your evaluation?


2. I am not convinced that the proposed approach would be effective in the presence of a large number of tasks or epochs (in the order of 100s). Can you please comment on the scalability of your approach? How would you expect the memory/compute requirements to scale with number of distinct elementary tasks in the "vocabularly"? What about the number of compositional tasks?

3. I think there is a highly relevant paper you do not cite: https://proceedings.neurips.cc/paper/2021/hash/fe5e7cb609bdbe6d62449d61849c38b0-Abstract.html
How does your probabilistic approach compare to their modular/gating-based technique?
I do not refer only to empirical results -- but also conceptually and in terms of scalability.

**Ethical Concerns:**

["NO or VERY MINOR ethics concerns only"]

**Limitations:**

For the most part the authors acknowledge the limitations of their study. One thing that is not clear however is the "maximum cognitive capacity" of their system -- in a scenario where the number of tasks keeps increasing with time while the memory/compute resources of the system are finite. How would the model behave in that case? Would it start forgetting in a "graceful/gradual" way (similar to an older person that gradually forgets tasks he/she has not practiced for a long time) or would it be something more unpredictable and abrupt?

**Quality:**

4

**Strengths And Weaknesses:**

Quality:
+  methodologically rigorous, offers clear theoretical formulation of context-modulated RNN architectures.
+ demonstrates solid empirical validation across standard cognitive neuroscience tasks.
- experiments primarily involve synthetic tasks -- questionable whether approach would work in more real-world tasks

Clarity
+ clearly defines two-system approach ("what" vs "how")
+ concepts & experiments are well-organized/presented


Significance:
+ significant advance in CL by modeling computational epochs and compositionality
+ illustrates backward and forward transfer, compositional generalization, task reuse -- goes further than prior work by Duncker et al.
- it would be better if the tasks also included something more realistic, from real-world applications that require CL using RNNs

Originality
- As expected, some conceptual overlap exists with previous works focusing on modular or context-dependent computations
+ but the originality comes from interpretable probabilistic model to guide modular reuse of computations in RNNs

---

> ### Author Rebuttal · Authors · 2025-07-30
>
> We thank the reviewer for their interest in our work and thoughtful comments. Concerns and questions raised in the review are addressed below.
> 1. **More realistic tasks.** At the present stage, we have chosen to focus on synthetic problems common in neuroscience modeling work for interpretability and relevance to understanding continual learning in the brain. The set of cognitive tasks has been of recent interest in the field (e.g., see refs. 11, 13, 14, 16, 35), and we followed this literature to use them as a testbed for our proposed architecture. However, we wholeheartedly agree with the reviewer that it is important to evaluate our approach on realistic time-series tasks and that NLP tasks would be particularly well-suited to the kind of compositionality we are concerned with here.
> 2. **Scalability** is indeed an important consideration for continual learning systems. In our case, letting $N_z$ denote the number of epochs (i.e., the ‘elementary tasks’), the probabilistic learning/inference algorithm (Sec. 3.3) used by the 'what' system would have time complexity $O(N_z^2)$ and space complexity $O(N_z)$, as is standard for learning/inference algorithms for HMM-like models. For the 'how' system, the main cost arises from running the RNN, which does not scale with $N_z$ (computing the effective weights in Eqs. 3-5 has cost linear in $N_z$). To improve scalability, a possible next step is to discourage the addition of new epochs by penalizing having a large $N_z$. Alternatively, it may also be possible to group epochs that are statistically distinct, but can share the same solution through another learning mechanism. This would be particularly relevant for the response epochs in our task set. It is also worth noting that, since each component of the network uses low-rank recurrent weights, the memory cost of adding a new component is linear in RNN size, as opposed to the quadratic scaling in vanilla RNNs.
> 3. **Related work.** We thank the reviewer for pointing us to the highly relevant Ostapenko et al., 2021. Our scheme differs from theirs in several conceptually important ways, as listed below. We have added reference to this work and incorporated a condensed version of this discussion into the 'Related work and our contributions' section (Sec. 2).
> 	1. Ostapenko et al. requires passing every datum to all modules and running the computation in all modules first to determine which module outputs to keep (their Eq. 5). 'Module selection' in our model (computing the effective weights in Eqs. 3-5) is determined by a separate inference algorithm that is independent of the RNN modules. Our proposed scheme is more efficient in settings where the number of modules is large and the cost of running each module is high.
> 	2. Since Ostapenko et al. deals with static inputs and feedforward networks, different components in the same layer can be run in parallel. In our case, tasks have temporal structures and are solved by recurrent networks. Importantly, the composition of network weights changes during each trial dynamically. Since the hidden state activity $\mathbf{h}_t$ is shared, different components thus interact with each other over time. Our approach allows for more flexible compositions of modules. For example, the same two modules can be used in the same trial in an alternating fashion.
> 	3. Finally, as the reviewer noted, our model provides an interpretable probabilistic description of the compositional structure of each task, which is more opaque in Ostapenko et al.
> 4. **Graceful forgetting.** This is an interesting question and not something our current learning algorithm is adapted to. We assume that the number of contexts across tasks does not exceed the capacity of the network (i.e. number of contexts $\times$ low-rank components per context $\le$ total dimensionality of the RNN), such that overwriting is not an issue. However, it would be possible to implement a mechanism that decays the association between context (what) and recurrent weights (how) over time, if a given context is not visited for a long duration. This would effectively free up the low-rank component to be overwritten and associated with a new context, and enable graceful forgetting. While this exceeds the scope of the current paper, it would be an interesting topic for future work.

---

> > ### Comment · Reviewer_fkcB · 2025-08-06
> >
> > I have read the other reviews and the rebuttals offered by the authors (btw, they were very well done and presented).
> >
> > It seems that even the two reviewers that were initially more concerned about the significance or novelty of the paper have now increased their scores.
> >
> > I will keep my score to "Accept" -- as I think that the paper is certainly worth publishing. I cannot give an even higher score because, as mentioned in the review, the paper mostly focuses on simple, synthetic tasks -- and so naturally there are some concerns about how relevant this framework would be in more realistic applications.

---

> > > ### Author Response · Authors · 2025-08-06
> > > **Thank you**
> > >
> > > We thank their reviewer for the appreciation of our work and our responses. Generalizing to more complex tasks is indeed an interesting next step and we are addressing it in our ongoing work.

---

### Note · Authors · 2025-08-13

We would like to thank the reviewers again for their thoughtful comments and actionable feedback. We addressed the raised concerns by clarifying the writing, adding further discussion of our work in relation to the CL in RNN literature, and performing additional baseline comparisons. We are glad to see that the reviewers appreciated the rebuttals, with reviewers jVxy and N8zU raising their scores.

In both machine learning and computational neuroscience, task compositionality (solving a task by breaking it down into subtasks) is a capability of increasing significance and interest. As the reviewers pointed out, our work provides a biologically motivated, conceptually novel and theoretically rigorous method for compositionality in RNNs. Extensive comparisons against existing methods demonstrates its advantage during continual learning of multiple related tasks, including mitigated forgetting, generalization to novel tasks, and forward/backward transfer. We believe that our work provides a theoretically tractable framework for task compositionality (Sec. 3) and solid empirical baseline (Sec. 4) for challenging CL problems with widely used neuroscience tasks. Thus, our work may appeal to a broad range of NeurIPS audience.

---

### Decision · Program_Chairs · 2025-09-17

**Decision:**

Accept (poster)

**Comment:**

The paper proposes an interesting method for solving tasks through compositional computation. It is based on building sequentially a vocabulary of tasks.  Then it utilizes a two-component model based on the so called "what-system" that identifies the current task context and the "how-system" that learns to perform the specific task. The whole system is based on RNNs and it is trained online. The authors demonstrate their ContextRNN method on neuroscience tasks and they show that it can outperform other techniques,  such as Elastic Weight Consolidation and Orthogonal Weight-Space Projection. More interestingly, the model can do forward and backward transfer to related tasks as well as compositional generalization.

All reviewers found the separation of "what" and "how" in continual learning of RNNs a very novel and interesting idea.
Overall, this is a strong submission with a clear and nice proposal as well as convincing experimental results.